# Distinct conformational states enable transglutaminase 2 to promote cancer cell survival versus cell death
Cody Aplin [1,2,4], Kara A. Zielinski [3,4], Suzette Pabit[3], Deborah Ogunribido[1], William P. Katt [2], Lois Pollack [3], Richard A. Cerione [1,2] ✉ & Shawn K. Milano[1,2]

Transglutaminase 2 (TG2) is a GTP-binding, protein-crosslinking enzyme that has been investigated as a therapeutic target for Celiac disease, neurological disorders, and aggressive cancers. TG2 has been suggested to adopt two conformational states that regulate its functions: a GTP-bound, closed conformation, and a calcium-bound, crosslinking-active open conformation. TG2 mutants that constitutively adopt an open conformation are cytotoxic to cancer cells. Thus, small molecules that bind and stabilize the open conformation of TG2 could offer a new therapeutic strategy. Here, we investigate TG2, using static and time-resolved small-angle X-ray scattering (SAXS) and single-particle cryoelectron microscopy (cryo-EM), to determine the conformational states responsible for conferring its biological effects. We also describe a newly developed TG2 inhibitor, LM11, that potently kills glioblastoma cells and use SAXS to investigate how LM11 affects the conformational states of TG2. Using SAXS and cryo-EM, we show that guanine nucleotides bind and stabilize a monomeric closed conformation while calcium binds to an open state that can form higher order oligomers. SAXS analysis suggests how a TG2 mutant that constitutively adopts the open state binds nucleotides through an alternative mechanism to wildtype TG2. Furthermore, we use time resolved SAXS to show that LM11 increases the ability of calcium to bind and stabilize an open conformation, which is not reversible by guanine nucleotides and is cytotoxic to cancer cells. Taken together, our findings demonstrate that the conformational dynamics of TG2 are more complex than previously suggested and highlight how conformational stabilization of TG2 by LM11 maintains TG2 in a cytotoxic conformational state.

Transglutaminase 2 (TG2) is the most ubiquitously expressed member of the calcium ($Ca^{2+}$) dependent transglutaminase family[1–3]. The primary function of TG2 is to cross link proteins via the formation of amide bonds between glutamine residues and primary amine groups. It has also been reported that TG2 can function as a GTPase[4,5], a protein disulfide isomerase[6], a kinase[7], and a DNA hydrolase[8], as well as a possible participant in GPCR-dependent signal transduction[9]. These diverse functions allow TG2 to play a role in numerous physiological processes, including cellular differentiation[10–15], cell adhesion[16,17], and endocytosis[18–20]. In neurodegenerative disorders, such as Alzheimer's and Huntington's disease, the crosslinking activity of TG2 has been suggested to play an important role in the pathogenesis of the disease and to contribute to the accumulation of

crosslinked protein aggregates[21–27]. TG2 is also one of the most important factors in Celiac disease, where it crosslinks gluten peptides to increase their immunoreactivity[28–31]. In glioblastoma, guanine nucleotide-binding by TG2 promotes glioma cell proliferation, survival, and invasion by interfering with the degradation of the EGF receptor (EGFR), resulting in increased EGFR signaling[3,15,32–36] and subsequent cell survival.

X-ray crystallography structures of TG2 have revealed that GDP or GTP binds to a "closed" conformational state that occludes the protein-crosslinking (transamidation) active site and thus eliminates its crosslinking activity (Fig. 1a)[37–40]. ATP also binds to the closed state of TG2, although with a much weaker binding affinity compared to GTP[41]. $Ca^{2+}$ binds and stabilizes the open conformational state of TG2, which disrupts the

[1]Department of Chemistry and Chemical Biology, Cornell University, 14853 Ithaca, NY, USA. [2]Department of Molecular Medicine, Cornell University, 14853 Ithaca, NY, USA. [3]School of Applied and Engineering Physics, Cornell University, 14853 Ithaca, NY, USA. [4]These authors contributed equally: Cody Aplin, Kara A. Zielinski. ✉e-mail: rac1@cornell.edu

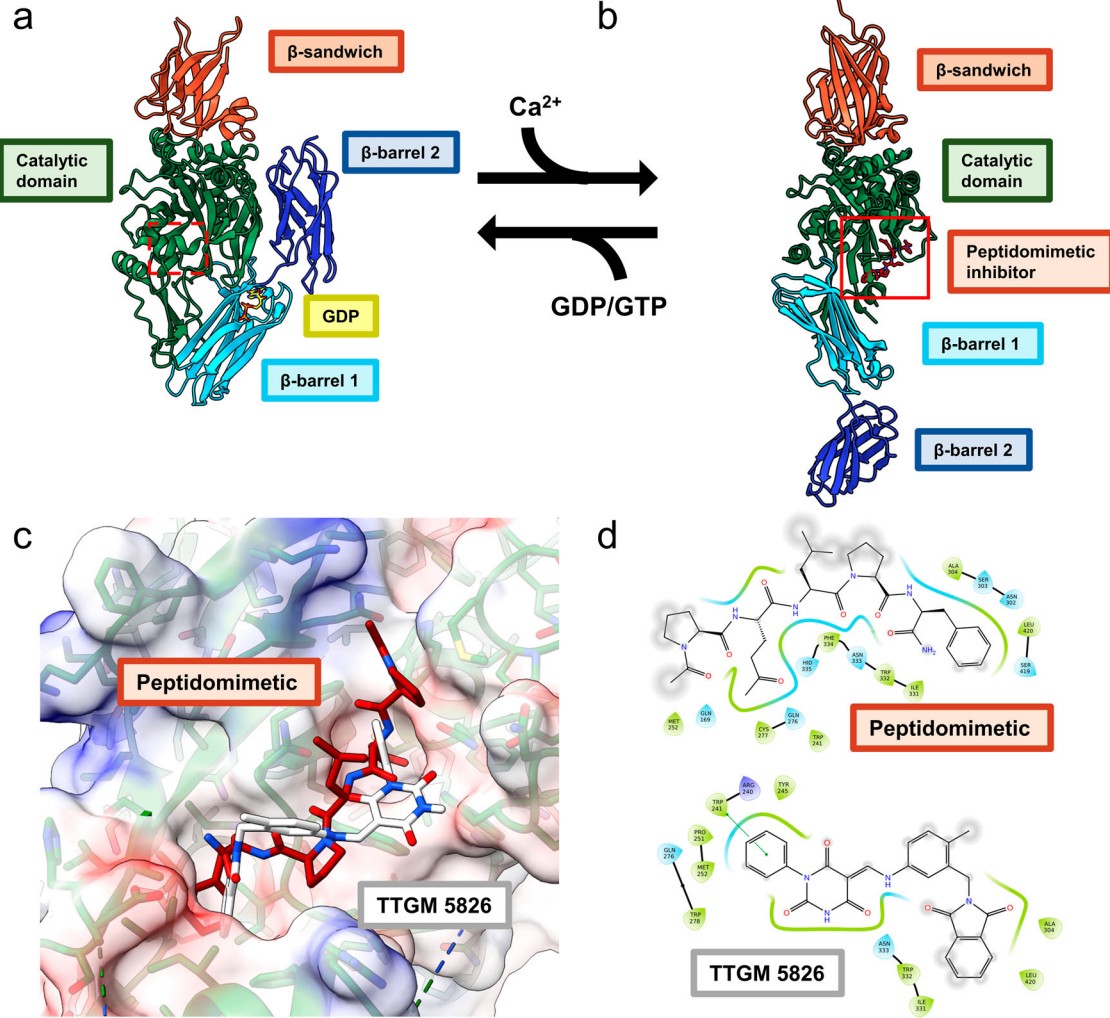

**Fig. 1 | Overview of TG2 conformational changes and the existing active site inhibitors. a** GDP or GTP bind to TG2 in a closed state (PDB: 1KV3, GDP shown in yellow). Calcium or active site inhibitors bind TG2 in an open conformation (**b**) (PDB: 2Q3Z, peptide inhibitor in maroon) that is crosslinking active. The drug binding site in (**c**) is highlighted with a solid box (red) to indicate it is accessible. Numerous studies have suggested that the closed state of TG2 promotes cell survival while the open state is cytotoxic through an unknown mechanism. The N-terminal β-sandwich domain is colored orange, the catalytic domain is colored green, and the β-barrels are colored cyan (β-barrel 1) and blue (β-barrel 2). **c** Docking pose of TTGM 5826 and a peptidomimetic inhibitor in the active site of TG2 (**d**) Ligand interaction diagrams of the peptidomimetic inhibitor (top) and TTGM5826 (bottom) with TG2.

nucleotide binding pocket, as observed in X-ray crystal structures of TG2 bound to peptidomimetic inhibitors (Fig. 1b)[33,42]. Moreover, the requirement for $Ca^{2+}$ when assaying TG2-catalyzed protein crosslinking led to the assumption that this divalent cation must be capable of stabilizing an open conformation to make the transamidation active site accessible to substrates. However, X-ray crystallography studies of the calcium bound form of TG2, and other transglutaminases have thus far reported only the closed conformation of TG2[43–45]. It is widely accepted that TG2 has multiple calcium binding sites, although the location and number of calcium binding sites have been disputed[42]. Despite binding to different sites on TG2, guanine nucleotides and $Ca^{2+}$ have been shown to allosterically regulate TG2 by differentially stabilizing the open and closed conformational states, respectively. In the cytoplasm, where the concentration of GTP (~0.5 mM)[46] is much greater than that of calcium (~0.1–1 µM)[47], TG2 is thought to bind to the closed conformation, whereas the calcium levels in the extracellular matrix (1–3 mM)[48] would be sufficient to stabilize the crosslinking-active open conformation. Interestingly, ectopic expression of TG2 mutants (i.e., R580K) that have a weakened affinity for guanine nucleotides are reported to adopt the open conformation and induce apoptosis independently of transamidation activity[49,50]. Thus, the ability of TG2 to assume different conformational states is critical to regulating its intracellular functions.

Due to its role in various diseases, several efforts have been undertaken to develop TG2 inhibitors[2,3,51,52]. These include peptidomimetic small molecules based on a gluten peptide, which TG2 binds tightly (Fig. 1c, d)[28–31], and several crystal structures of TG2 in the open state have been solved by occupying the transamidation active site with a peptidomimetic inhibitor[53]. We were interested in developing a new class of specific, small-molecule inhibitors of TG2 that were able to maintain its cytotoxic, open conformation within cancer cells, and led to the discovery of TTGM 5826 (Fig. 1c, d)[54]. We demonstrated that TTGM 5826 was able to inhibit the protein crosslinking activity of TG2, maintain it in an open conformation, and prevent the growth of TG2-expressing glioblastoma cell lines and glioma stem cells. However, we were left with one key question: if intracellular TG2 predominantly adopts the closed conformation, how was TTGM 5826 stabilizing the open state to kill cancer cells? Answering this question would aid in the development of new inhibitors and help identify the biological mechanism through which open state TG2 induces cytotoxicity.

To robustly characterize the conformational dynamics that TG2 undergoes, we used cryoelectron microscopy (cryo-EM) and small angle X-ray scattering (SAXS) to study the effects of guanine nucleotides and calcium on its structure[55–57]. Cryo-EM has evolved to become an important

method for experimental protein structure determination and enables high-resolution reconstruction without the need for a protein crystal. However, cryo-EM has faced significant challenges in achieving high-resolution structures of small proteins and proteins with flexible domains. At 77 kDa, TG2 is a relatively small protein for cryo-EM, and the highly dynamic nature of its conformational changes provides an additional complication. X-ray crystallography has been used to obtain 3D structures for TG2; although the constraints imposed by formation of a crystal lattice may mask the conformational and oligomeric transitions that TG2 undergoes in response to different ligands and inhibitors[33,37–39,41,42,58]. This prompted us to utilize small angle X-ray scattering (SAXS), as this method is performed in solution and at room temperature making it well suited to study the conformational dynamics of flexible proteins and to distinguish between the open and closed forms of TG2[59–61]. Additionally, SAXS can be combined with time-resolved techniques to study the dynamics of such structural transitions. In particular, stop-flow[62], diffusive[63], turbulent[64], and chaotic advection[65,66] mixers have all been successfully coupled to SAXS. These approaches rapidly mix two molecules together and then capture snapshots at different timepoints to understand the mechanism of a reaction.

Here, we use a combination of high-resolution cryo-EM structures and low-resolution static and time-resolved SAXS to show how ligand binding influences the conformational and oligomeric states of TG2, demonstrating that TG2 undergoes more complex structural transitions than previously understood. We also describe the first round of optimization efforts aimed at finding small molecules with improved potency and solubility compared to TTGM 5826 and demonstrate that our best molecule of this class to date, LM11, is more potent than TTGM 5826 in triggering cell death in aggressive glioblastoma cell lines. Furthermore, we use SAXS to examine how LM11 affects the structural dynamics of TG2. We show that its mode of action involves enhancing the ability of calcium to bind to TG2 and stabilize an open conformational state that cannot be reversed even at high concentrations of guanine nucleotide, which could have important ramifications for the further development of therapeutic compounds that effectively target intracellular TG2 in cancer.

## Results
### Characterization of TG2 conformational states stabilized upon the binding of guanine nucleotides

To improve our understanding of how the conformational dynamics of TG2 are impacted by the binding of guanine nucleotides, we performed SAXS titration series experiments to examine the interactions of GTP with wildtype TG2 and a cytotoxic TG2 mutant. We first investigated the transition of recombinant human TG2 from its purified state to the GTP-bound state by titrating in 0 to 5 mM GTP. As shown by the Kratky plot in Fig. 2a, which shows SAXS intensity as a function of scattering angle and provides information on particle size, shape, and flexibility, increasing additions of GTP increased the amount of wildtype TG2 in a compact conformational state, with a Guinier radius of gyration ($R_g$) of 32 Å (Fig. 2b, Table S1). At 0.5 mM of guanine nucleotides and higher, the scattering profile can be well-modeled by the crystal structure of guanine nucleotide bound monomeric TG2 in the closed state (PDB: 4PYG) (Fig. 2c, d). As calculated using the FoXS server[67], GTP-bound monomeric closed state TG2 has a theoretical $R_g$ of 29.2 Å and maximum dimension ($D_{max}$) of 103 Å which is consistent with the P(r) distribution of TG2 treated with 5 mM GTP (Fig. S1a). Previous crystallographic studies of TG2 bound to GDP have shown that TG2 has a propensity to crystallize as a closed state dimer (Fig. S2)[37]. To confirm that the structure of TG2 bound to guanine nucleotide is in fact a closed state monomer, we used cryo-EM to solve a 3.2 Å structure of TG2 bound to GDP (Fig. 2e, Table 1). The cryo-EM structure of GDP-bound TG2 is in excellent agreement with the crystal structure with an overall RMSD of 0.8 Å; TG2 adopts the monomeric closed state and the cryo-EM map displays strong density for the nucleotide binding site and the bound GDP molecule (Fig. 2e inset)[37]. As described in crystallographic studies, GDP is stabilized through π-π stacking interactions of the guanine ring with Phe174 as well as several

hydrogen bond interactions between the α and β phosphate of GDP and a trio of arginine residues, Arg476, Arg478, and Arg580.

Previous studies have indicated that TG2 mutants in which Arg580 has been changed have a significantly weakened affinity for GTP [48] and assume a cytotoxic open conformation[50,68]. However, when we examined the response of TG2 R580K to increasing concentrations of GTP up to 5 mM (Fig. 3a), surprisingly, we found that the R580K mutant responded to high concentrations of GTP, albeit with an apparent lower affinity for GTP compared to wildtype TG2 (Fig. 3b). Unlike wildtype TG2, the Guinier Rg value suggests that GTP is not binding to the R580K mutant in a closed conformational state. Rather, the TG2 R580K mutant, in the absence of added nucleotide or Ca²⁺, shows an upturn at low q in the Kratky plot (Fig. 3a) suggesting it forms a higher order oligomeric species which then steadily decreases with increasing GTP concentrations, ultimately resulting in a conformational state with a Guinier Rg value much higher than that for wildtype TG2 bound to GTP. Previous SAXS studies of the TG2 R580K mutant, which has been suggested to constitutively adopt the open state, showed that it forms a dimer in solution[56]. Upon inspection of the crystal structure of the open state monomer (PDB: 2Q3Z), the symmetry packed molecules assemble into an open state dimer, as shown in Fig. 3c. At GTP concentrations greater than 1 mM, the scattering profile of the R580K mutant can be fit well to the open state dimer (Fig. 3d). To investigate a potential model for how open state TG2 may be able to bind nucleotides, we aligned the closed state monomer structure to the open state dimer using the catalytic domain of TG2 (Fig. 3e). Intriguingly, when the closed state monomer is aligned to the catalytic domain of one open state TG2 monomer within the open state dimer, the β-barrel 2 domain of the other open state monomer also aligns with the closed state monomer, suggesting that the R580K mutant may be capable of binding guanine nucleotides through contributions of both monomers within the dimer, resulting in the stabilization of the open state dimer conformation.

### Characterization of TG2 conformational states stabilized upon the binding of Ca²⁺

It has been previously shown that Ca²⁺ stabilizes an open conformational state for TG2 that is no longer capable of binding guanine nucleotides[4,42]. The interactions of GTP and Ca²⁺ with TG2 have appeared to be competitive as each ligand weakens the affinity of the other for the protein, despite having different predicted binding sites; thus, these effects are apparently due to allosteric regulation rather than to direct binding competition. To better characterize Ca²⁺ binding to TG2, we examined the interactions of Ca²⁺ with TG2 by cryo-EM (Fig. 4a). The resulting cryo-EM map contains density only for the N-terminal β-sandwich domain and the catalytic domain of TG2, suggesting that the C-terminal β-barrel domains are highly flexible. While the gold-standard FSC reported in cryoSPARC is 3.4 Å, analysis of the cryo-EM map with 3DFSC indicates the map is somewhat anisotropic (Sphericity = 0.68), resulting in a global resolution of 4.2 Å which is insufficient for generating an ab initio atomic model[69]. However, given the existence of previously solved X-ray crystal structures of a closely related transglutaminase family member, TG3, bound to Ca²⁺ and its similarity to TG2 (38.7% sequence identity with three conserved Ca²⁺ binding sites), we used Prime (Schrödinger, Inc) to generate a homology model of TG2 bound to Ca²⁺ at the three conserved binding sites and found that it was in good agreement with the cryo-EM map (Fig. 4b, Table S2)[43,70,71]. At each of the three Ca²⁺ binding sites, the atomic model of TG2 bound to GDP is a poor fit to the cryo-EM map, while the refined homology model generated using TG3 is an excellent fit (Figs. S3a–c).

Due to the apparent flexibility of the C-terminal β-barrel domains, as evidenced by missing density in the cryo-EM map, we used SAXS to attempt to model the full-length protein. We found that when human TG2 was treated with 2 mM Ca²⁺ in solution, the scattering profile displayed a shape change suggesting an elongation and increase in the size of TG2 (Fig. 4c), which was supported by the increase in the Guinier Rg and the forward scattering, I(0), and is indicative of an oligomeric species (Table S1). In addition, the data is a poor fit to the open state monomer crystal structure

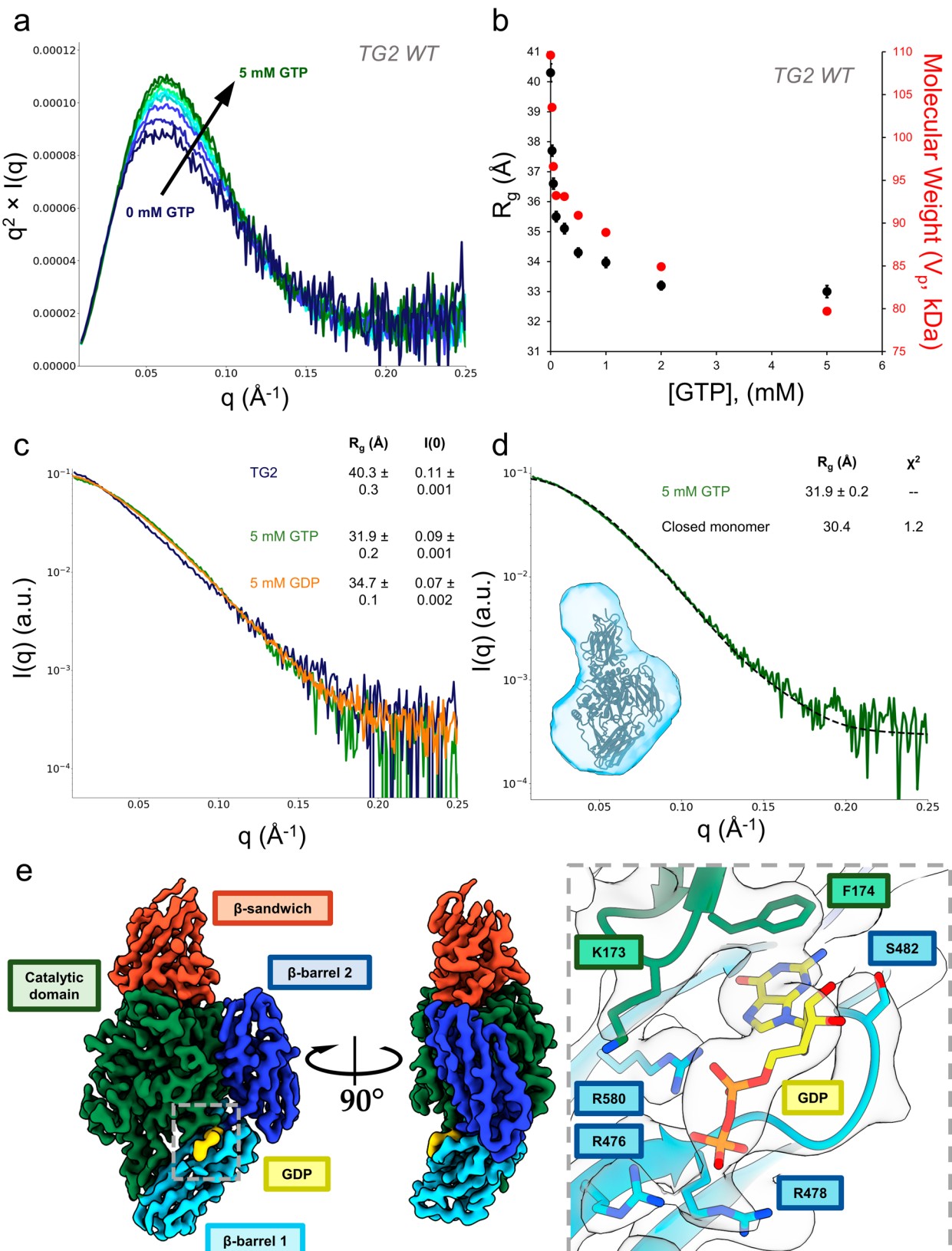

solved for TG2[53]. Reconstruction of the SAXS envelope was performed either using no symmetry (P1) or with a twofold symmetry axis (P2), and both envelopes are consistent with an elongated TG2 dimer (Figs. S4c, d). Interestingly, we found that the open state dimer that is formed by the TG2 R580K mutant in response to nucleotide binding nicely fits the scattering profile of TG2 treated with Ca$^{2+}$ (Fig. 4c). Taken together, our data suggests that when TG2 adopts the open state, either by the introduction of a cytotoxic R580K substitution which weakens the affinity for guanine nucleotides, or when stabilized upon the binding of Ca$^{2+}$, it has a propensity to dimerize. Our experiments also suggest that the ability of TG2 to dimerize

**Fig. 2 | SAXS and cryo-EM demonstrates that guanine nucleotides bind to human TG2 in the closed state. a, b** Titration of 0 to 5 mM GTP into wildtype TG2 (TG2 WT). **a** The Kratky plot of TG2 for the titration series from 0 mM (dark blue) to 5 mM GTP (dark green). **b** The Guinier $R_g$ decreases as TG2 adopts a compacted, saturated conformational state. The experimental molecular weight calculated from the Porod Volume is shown on the secondary $Y$-axis (red). **c** The scattering profiles of human TG2 expressed in *E. coli* (dark blue) undergo a similar change when treated with 5 mM GDP (orange) or 5 mM GTP (green). The Guinier $R_g$ and I(0) are shown

in the legend. **d** Nucleotide bound TG2 can be described as a closed state monomer in solution, as demonstrated using CRYSOL to fit to the crystal structure of TG2 bound to GTP (black, PDB ID: 4PYG). The SAXS envelope is shown in light blue. The Guinier $R_g$ and the model $X^2$ are shown in the legend. **e** The 3.2 Å cryo-EM structure of TG2 bound to GDP confirms that GDP bound TG2 is monomeric and adopts the closed state. The cryo-EM map is colored by TG2 domain and map density for the GDP binding site within is shown in the inset.

## Table 1 | Cryo-EM data collection, refinement, and validation statistics

| | TG2 bound to GDP (EMD-41574) (PDB 8TR9) | TG2 bound to calcium (EMD-42356) |
|---|---|---|
| **Data collection and processing** | | |
| Magnification | 79,000 | 79,000 |
| Voltage (kV) | 200 | 200 |
| Electron exposure (e–/Å$^2$) | 50 | 50 |
| Defocus range (μm) | −1.2 to −2.4 | −1.2 to −2.4 |
| Pixel size (Å) | 1.034 (0.517 super-res) | 1.034 (0.517 super-res) |
| Symmetry imposed | C1 | C1 |
| Initial particle images (no.) | 748,598 | 1,400,060 |
| Final particle images (no.) | 136,297 | 68,975 |
| Map resolution (Å) FSC threshold | 3.20 0.143 | 3.44 0.143 |
| Map resolution range (Å) | 2.9–4.8 | 3.1–20.2 |
| **Refinement** | | |
| Initial model used | ModelAngelo | |
| Model resolution (Å) FSC threshold | 3.20 0.143 | |
| Map sharpening *B* factor (Å$^2$) | 110 | |
| **Model composition** | | |
| Non-hydrogen atoms | 5442 | |
| Protein residues | 684 | |
| Ligands | 1 | |
| **B factors (Å$^2$)** | | |
| Protein | 98.5 | |
| Ligand | 95.6 | |
| **R.m.s. deviations** | | |
| Bond lengths (Å) | 0.002 | |
| Bond angles (°) | 0.542 | |
| **Validation** | | |
| MolProbity score | 1.75 | |
| Clashscore | 9 | |
| Poor rotamers (%) | 0.00 | |
| **Ramachandran plot** | | |
| Favored (%) | 100 | |
| Allowed (%) | 0.00 | |
| Disallowed (%) | 0.00 | |

is dependent on protein concentration. The SAXS experiments were carried out at 25 μM TG2 to achieve a sufficient scattering signal, while the cryo-EM analysis was carried out using 3 μM TG2 as was necessary to obtain an appropriate particle density, which made it possible to capture the monomeric form of the TG2 open state. A SAXS concentration series and SEC-MALS experiment for TG2 in storage buffer were performed to further demonstrate the concentration dependence on dimerization (Fig. S5).

We next determined whether we could monitor the competition between the binding of $Ca^{2+}$ and GTP to TG2 and the effects on its conformational states using SAXS. First, we incubated 25 μM TG2 with 2 mM $CaCl_2$ for five minutes and then added 5 mM GTP. The scattering profile indicates the effects on TG2 structure that accompany the binding of $Ca^{2+}$ can be reversed by GTP, as the Guinier $R_g$ value is consistent with TG2 forming a closed state monomer (Fig. 4d). In the presence of GTP, increasing concentrations of $Ca^{2+}$ results in a steady increase in the Rg values, indicative of $Ca^{2+}$ exerting a negative regulatory effect on GTP binding (Fig. 4e), as previously observed. When a similar $Ca^{2+}$ titration was performed with the TG2 R580K mutant, we found that it transitions from the dimeric state to a higher order oligomeric species (Fig. 4f). Taken together, our SAXS results show that $Ca^{2+}$ and GTP can reversibly modulate the conformational ensemble of TG2 and that the TG2 R580K mutant forms a higher order oligomer at lower $Ca^{2+}$ concentrations compared to wildtype TG2.

### The development of improved conformational state TG2 inhibitors

We originally identified TTGM 5826 as a candidate small molecule that might help to trap TG2 in an open state and thereby induce a cytotoxic effect in cancer cells. In fact, we found that TTGM 5826 in the presence of $Ca^{2+}$ maintained TG2 in an open state conformation and inhibited its protein crosslinking activity[54]. To identify more potent analogs of TTGM 5826, we examined commercially available small molecules that were similar in chemical structure to TTGM 5826 (the LM-series, Fig. S6). We first tested their ability to alter the conformation of TG2 by assaying the binding of the fluorescent GTP analog, BODIPY-GTP, to TG2. BODIPY-GTP binds in the guanine nucleotide binding site of TG2, resulting in an increase in BODIPY fluorescence emission, and stabilizes the closed state conformation whereas the addition of $Ca^{2+}$ helps to maintain TG2 in the open state. This results in the dissociation of BODIPY-GTP as a read-out by a decrease in BODIPY fluorescence emission. TG2 (150 nM) and BODIPY-GTP (500 nM) were incubated with each of the LM compounds (50 μM), and the fluorescence emission of BODIPY-GTP was monitored upon subsequent additions of 10 mM $Ca^{2+}$. Each drug was also screened for its ability to inhibit TG2 crosslinking activity. We found that while several of the compounds demonstrated some ability to stabilize the open state of TG2 as indicated by the BODIPY-GTP binding assays, only LM11 was more potent at inhibiting TG2 crosslinking activity compared to TTGM 5826 (Fig. 5a (Full gel image in S7a) and Table S3). Docking analysis using Glide in Maestro (Schrödinger, Inc) suggested that LM11 could have improved binding to TG2, with one chlorine atom replacing a hydrogen atom to project into a narrow cleft and interact with Tyr245, and a second chlorine replacing a solvent exposed methyl group projecting into water, possibly improving the interaction of the molecule with bulk water outside the binding site (Fig. 5b)[72]. QikProp also predicted that LM11 had a higher Caco2 permeability (predictive of oral

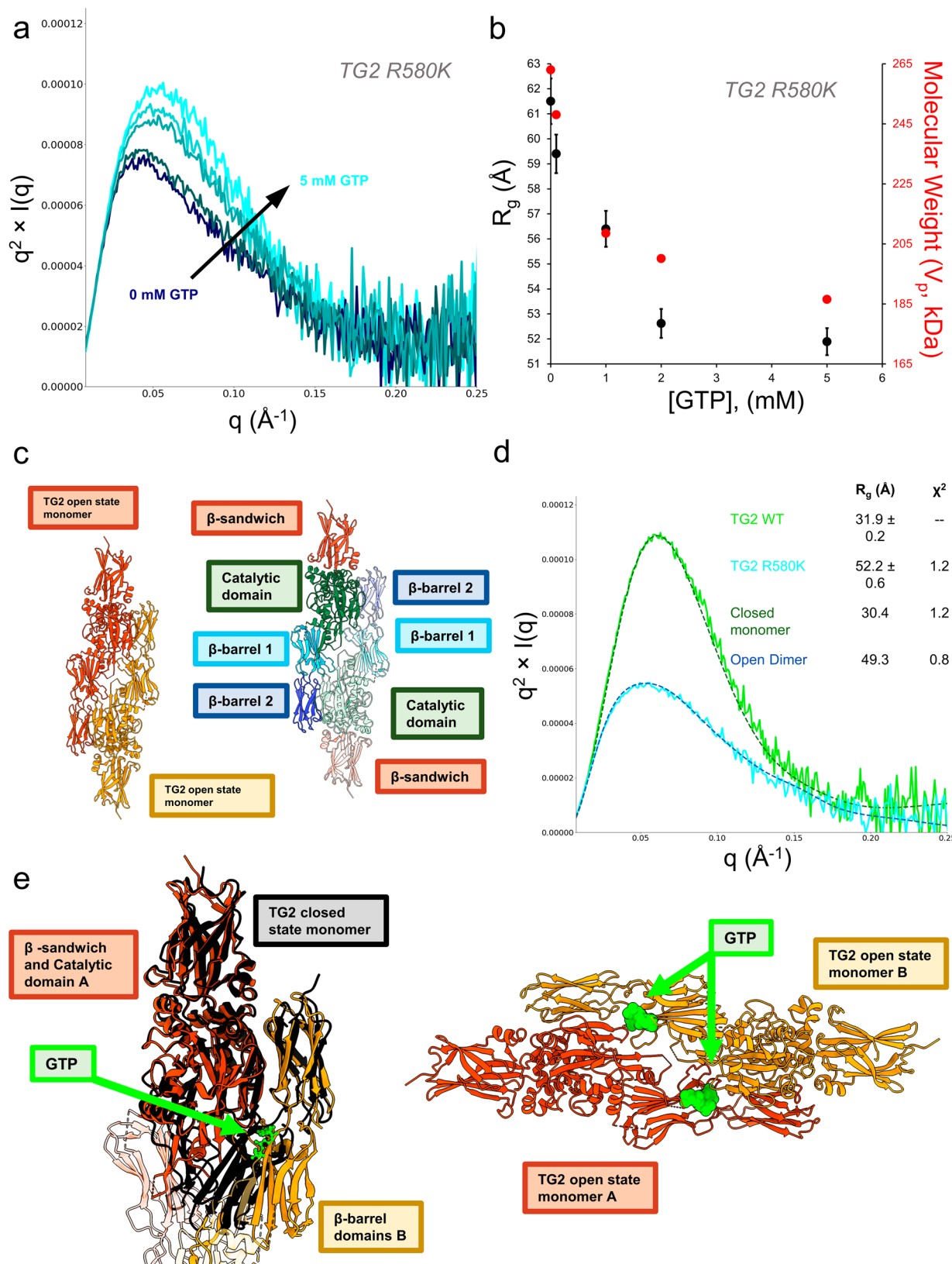

bioavailability) and MDCK cell permeability (predictive of ability to cross blood/brain barrier) in comparison to TTGM 5826, suggesting it may be an improved scaffold to develop orally available drugs (Table S3).

We next compared the effects of LM11 versus TTGM 5826 on the viability of glioma cells, using two different cell lines that expressed varying levels of TG2, namely U87-MG cells (U87) which highly express TG2, and T98G cells that express lower levels of TG2 (Fig. 5c, Full gel image in Fig. S7b). The cells were treated with compounds at concentrations between 1.5 and 25 μM and the percentage of dead cells after two days of treatment was determined with trypan blue staining. We found that LM11 more

**Fig. 3 | SAXS demonstrates that nucleotide binding defective TG2 mutants that constitutively adopt the open state dimerize in the presence of GTP. a, b** Titration of 0 mM to 5 mM GTP into TG2 R580K. **a** Kratky curves for the titration series are shown in shades of blue (dark blue, 0 mM to light blue, 5 mM). **b** The Guinier $R_g$ decreases as the conformational ensemble of TG2 R580K is driven to adopt a saturated conformational state. The experimental molecular weight calculated from the Porod Volume is shown on the secondary $Y$-axis (red). **c** The open state dimer generated from the crystal packing of the open state crystal structure of TG2. Left, the open state dimer colored by monomer. Right, the open state dimer colored by

domain. **d** Nucleotide bound TG2 R580K (light blue) can be described as an open state dimer in solution, as demonstrated by CRYSOL fit of the open state dimer (dark blue). Wildtype TG2 bound to GTP (light green) and the CRYSOL fit to the closed state monomer (dark green) are shown for comparison. The Guinier $R_g$ and the model $X^2$ are shown in the legend. **e** Left, the two monomers within the open state dimer (orange and yellow) form a highly similar GTP binding pocket to the closed state monomer (black). Right, proposed nucleotide binding sites within the open state dimer.

potently killed U87-MG cells compared to T98G cells, as its effectiveness correlated with TG2 expression (Fig. 5c, d). In addition, LM11 exhibited a lower $IC_{50}$ compared to TTGM 5826 in each of the glioma cells, indicating that it is a more potent inhibitor. In U87 cells, which highly express TG2, LM11 had a seven-fold increase in potency relative to TTGM 5826, with $IC_{50}$ values of 2.8 μM and 20.6 μM, respectively. In T98G cells, which express less TG2 than U87 cells, LM11 maintained a three-fold increase in potency relative to TTGM 5826, with $IC_{50}$ values of 7.4 μM and 20.8 μM, respectively.

### SAXS analysis of the interactions of LM11 with TG2

We then performed SAXS experiments to investigate how LM11 binding impacts the conformational state of TG2. In the absence of $Ca^{2+}$, neither the carrier agent DMSO, nor 50 μM TTGM 5826, or 50 μM LM11, had any effect on the conformational state of TG2 (Fig. S8). In the presence of $Ca^{2+}$ (a 5 min pre-incubation) and 50 μM LM11 (a 30 min incubation following the $Ca^{2+}$ pre-incubation), both the Rg value and I(0) increased when compared to TG2 exposed to $Ca^{2+}$ alone (Fig. 5e). In addition, a slight upturn at low q suggests the formation of a higher-order oligomer. Together, these findings suggest that inhibitor binding increases the amount of elongated, oligomeric TG2 species. We then tested whether GTP can reverse the effect of $Ca^{2+}$ in the presence of the inhibitors. Wildtype TG2 was again incubated with $Ca^{2+}$ (5 min) and LM11 (30 min) before the addition of 5 mM GTP. We found that in the presence of $Ca^{2+}$ and LM11, TG2 assumed a conformational state that was not reversible by GTP (Fig. 5e). This data suggests that as was proposed for TTGM 5826[54], LM11 acts as a conformational "door stop" which prevents TG2 from adopting a closed state. We attempted to model the LM11 bound species, using the same approaches as described above. The SAXS envelope determined with DAMMIF indicated an elongated structure ($D_{max}$ = 200 Å), similar to TG2 in the presence of $Ca^{2+}$ alone. In contrast, CRYSOL yielded a poor fit for the open state dimer when TG2 was treated with $Ca^{2+}$, LM11, and GTP (Fig. 5f). Similarly, OLIGOMER gave poor fits for the open state dimer and the closed state monomer. This suggests that the conformation adopted by TG2 in the presence of LM11 and $Ca^{2+}$ is distinct from that for TG2 bound to $Ca^{2+}$ alone and it cannot be reversed to a monomeric closed state by the addition of GTP.

### Time-resolved SAXS demonstrates that LM11 reduces the levels of $Ca^{2+}$ required to stabilize TG2 in an open state

To gain a better mechanistic understanding of how LM11 and $Ca^{2+}$ stabilize TG2 in an irreversible open conformation, we used time-resolved SAXS (TR-SAXS). A 3D printed Kenics-style chaotic advection mixer[66] was used to rapidly combine two fluids for reaction initiation (Fig. 6a), with the flow rates and position of the X-ray beam adjusted to facilitate timepoints ranging from 32–631 ms. The fully mixed species exits the Kenics into a flow cell and is surrounded by a sheath flow to reduce radiation damage and prevent the sample from adhering to the wall[73]. Importantly, the concentration of the ligand ($Ca^{2+}$ or LM11) can be kept relatively constant after mixing when it is introduced into the sheath flow as this prevents appreciable diffusion in the radial direction.

Using the Kenics, we captured the initial stages of the binding of $Ca^{2+}$ to TG2 in the absence and presence of LM11 (Fig. 6b, Tables S4, S5). The first time series (32, 63, 100, 316, and 631 ms) was obtained by mixing TG2 with $Ca^{2+}$ alone (originally published in[66], SASDRE3-SASDRP3). In the absence of LM11, $Ca^{2+}$ binding to TG2 was accompanied by a slow conformational

change for approximately 100 ms with an Rg of 40.7 ± 0.4 Å (Fig. 6b, black circles). TG2 then undergoes a slight elongation to 45.1 ± 0.6 by 316 ms that is maintained until 631 ms (Rg = 45.8 ± 0.5 Å). Further elongation of TG2 in the presence of $Ca^{2+}$ occurred on longer time scales, with a final Rg of 52.5 ± 0.5 Å after five minutes, as we observed with static SAXS. To visualize the transition from monomer to dimer, we then performed multi-state modeling with an in-house deconvolution using the closed state monomer and open state dimer (Fig. S9). The scattering profiles from 0–631 ms can be described by linear combinations of the closed state monomer and open state dimer, indicating that the relative amount of dimer increased from an initial state ($t$ = 0 ms) consisting of 39% monomer and 61% dimer to 100% dimer after five minutes (Fig. S10a). At even longer time scales, i.e., up to 30 min, the Porod volume of TG2 continually increases in the presence of $Ca^{2+}$, indicating the formation of a higher order oligomeric species (Fig. S10b).

A second time series (100, 316, and 631 ms) was captured by mixing TG2 with $Ca^{2+}$ and LM11 simultaneously (Fig. 6b, red circles). When LM11 was included, the response to calcium occurred more rapidly, with an elongated Rg of 45.2 ± 0.7 at 100 ms, which further increased to 46.8 ± 0.7 Å at 316 ms and then reached 50.9 ± 1.0 by 631 ms. This data further demonstrates that LM11 aids in stabilizing the open state of TG2 and suggests that this may be due to increasing the rate at which $Ca^{2+}$ binds to the protein. For the time series including LM11 and $Ca^{2+}$, we found that multi-state modeling using a combination of TG2 closed-state monomer and open-state dimer was unable to effectively describe the scattering profiles at any of the time points. The inability to fit the scattering profiles further suggests that the conformational state of TG2 bound to $Ca^{2+}$ is distinct from that for TG2 in the presence of both $Ca^{2+}$ and LM11.

To confirm that the presence of LM11 decreases the amount of $Ca^{2+}$ necessary to stabilize the open-state conformation of TG2, we performed static SAXS experiments in which TG2 and LM11 were premixed, and $Ca^{2+}$ were titrated in increasing concentrations (Fig. 6c). In the absence of added LM11, when TG2 was treated with up to 0.5 mM $Ca^{2+}$, it remained in the same conformation as that observed in the absence of added ligands, with an Rg of 42.8 ± 1.0 Å. However, in the presence of LM11, 0.5 mM $Ca^{2+}$ was sufficient to stabilize a conformational state in TG2 that yielded an Rg of 49.1 ± 0.7 Å, thus indicating that this small molecule inhibitor effectively increased the potency of $Ca^{2+}$. This mirrors the results obtained in the BODIPY-GTP fluorescence experiments in Fig. 5a above, where the presence of LM11 increased the effectiveness of $Ca^{2+}$ to reduce the binding of BODIPY-GTP to TG2.

### Discussion

In this work, we performed a structural characterization of TG2 to determine how small molecule inhibitors may be able to stabilize the open state of TG2 inside of cancer cells, despite the GTP-rich cellular environment. Using SAXS and cryo-EM, we comprehensively characterized the allosteric regulation of TG2 by GTP and $Ca^{2+}$, demonstrating that they can functionally compete with one another by binding to distinct conformational and oligomeric states (Fig. 7a). In addition, we demonstrated that the cytotoxic TG2 mutant, TG2 R580K, forms a GTP-bound open state dimer that is unable to adopt a monomeric, GTP bound state and oligomerizes at lower $Ca^{2+}$ concentrations than wildtype TG2. Furthermore, we identified a more potent small molecule inhibitor of TG2, LM11, and used time-resolved SAXS to show that LM11 reduces the rate and dosage required for $Ca^{2+}$ to bind to TG2 and stabilize its open and oligomeric states (Fig. 7b).

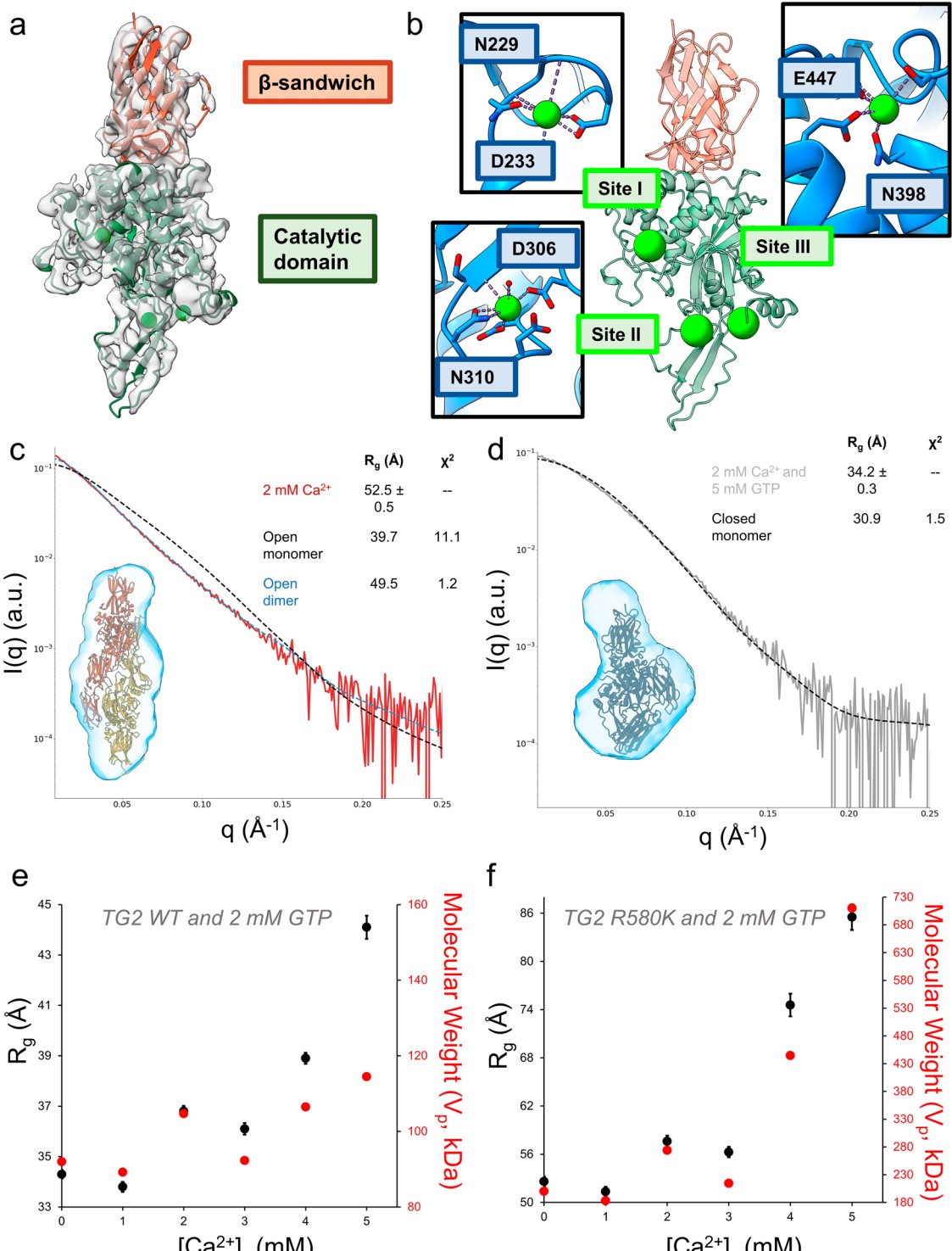

**Fig. 4 | SAXS and cryo-EM demonstrates that human TG2 adopts the open state when bound to $Ca^{2+}$ and forms a dimer at high enzyme concentrations. a** The 3.4 Å cryo-EM structure of the TG2 catalytic domain bound to $Ca^{2+}$ ions suggests that TG2 adopts the open state. **b** The three $Ca^{2+}$ binding sites (lime) within the TG2 catalytic domain. The conserved residues involved in $Ca^{2+}$ binding is shown in the inset for each site. **c** $Ca^{2+}$ bound TG2 can be described as a open state dimer in solution, as demonstrated using DAMMIF to construct a SAXS envelope (light blue) and CRYSOL to fit to the open state dimer (yellow, see Fig. 3c). The CRYSOL fit of the open state monomer is shown in black for comparison. The Guinier $R_g$ and the model $X^2$ are shown in the legend. **d** TG2 incubated with 2 mM $Ca^{2+}$ and then treated with 5 mM GTP (gray) can be described as a closed state monomer in solution, as

demonstrated using DAMMIF to construct a SAXS envelope (light blue) and CRYSOL to fit to the closed state monomer (black). The Guinier $R_g$ and the model $X^2$ are shown in the legend. **e** The Guinier $R_g$ of TG2 pre-incubated with 2 mM GTP increases in response to treatment with 1 mM to 5 mM $Ca^{2+}$, supporting the formation of the open state. The experimental molecular weight calculated from the Porod Volume is shown on the secondary $Y$-axis (red). **f** The Guinier $R_g$ of TG2 R580K pre-incubated with 2 mM GTP increases in response to treatment with 1 mM to 5 mM $Ca^{2+}$, indicating it can promote oligomerization of the R580K open state dimer. The experimental molecular weight calculated from the Porod Volume is shown on the secondary $Y$-axis (red).

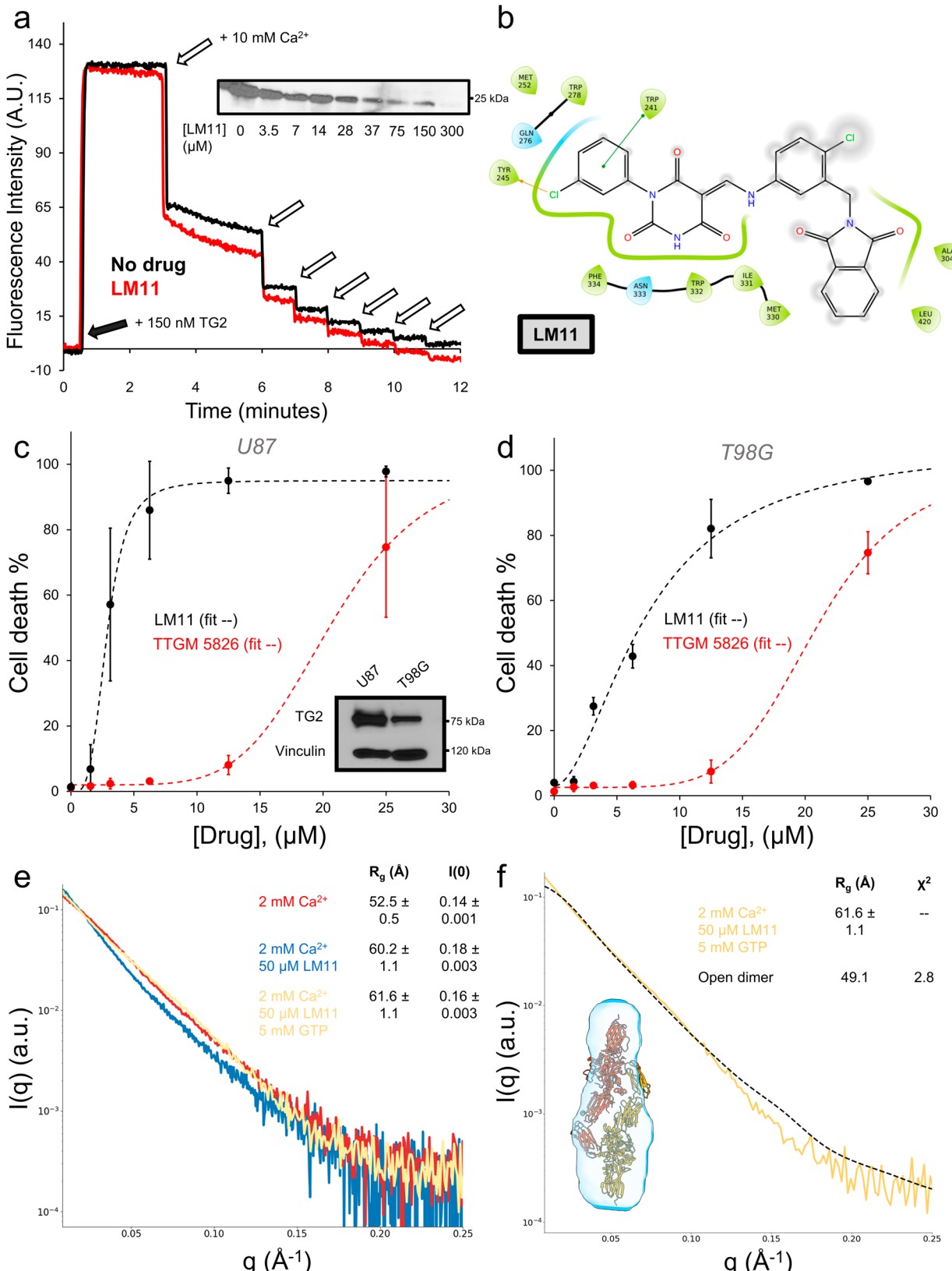

**Fig. 5 | LM11 is an improved TG2 inhibitor and stabilizes an elongated TG2 conformation. a** LM11 disrupts BODIPY-GTP binding by TG2 and inhibits TG2 crosslinking activity (inset). **b** Ligand interaction diagram of LM11 with TG2. **c, d** Dose dependent potency of the lead compound, TTGM 5826, and the improved inhibitor, LM11 in glioma cell lines expressing different levels of TG2. TG2 expression levels are shown as an inset in panel **c**. Data points are the average of three biological replicates and error bars are shown as the standard deviation ($n = 3$). **e** The scattering profiles of TG2 treated with 2 mM Ca$^{2+}$ (red), 2 mM Ca$^{2+}$ and 50 μM LM11 (blue), and 2 mM Ca$^{2+}$, 50 μM LM11, and 5 mM GTP (yellow). The Guinier $R_g$ and I(0) are shown in the legend. **f** The open state dimer does not fit the scattering profile of TG2 incubated with 2 mM Ca$^{2+}$, 50 μM LM11, and 5 mM GTP (yellow) as demonstrated using DAMMIF to construct a SAXS envelope (light blue) and CRYSOL to fit to the open state dimer (black). The Guinier $R_g$ and the model $X^2$ are shown in the legend.

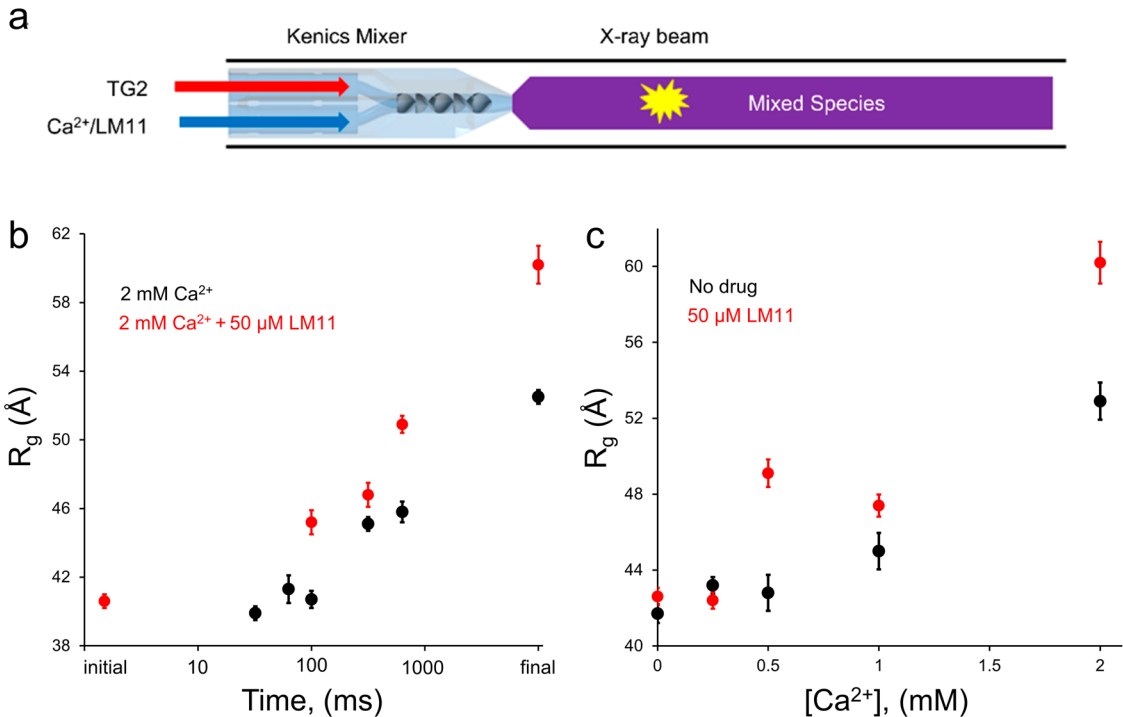

**Fig. 6 | Time-resolved SAXS of TG2 with Ca²⁺ and LM11. a** Schematic of sample flow in the Kenics mixer. **b** The time-resolved Guinier $R_g$ of TG2 mixed with 2 mM Ca²⁺ (black) increases faster in the presence of 50 μM LM11 (red). **c** The Guinier $R_g$ of TG2 mixed with Ca²⁺ in the absence (black) or presence of 50 μM LM11 (red) suggests LM11 increases the sensitivity of TG2 to calcium.

Importantly, we found that LM11 prevents GTP from reversing the conformational transitions adopted by TG2 in the presence of Ca²⁺, resulting in cytotoxicity and demonstrating a functional parallel with the TG2 R580K mutant which assumes a cytotoxic, oligomeric state that cannot be reduced to a monomeric species by GTP.

Through our comprehensive structural analyses, we identified a novel binding interaction between GTP and TG2, by studying the effect of GTP on the TG2 R580K mutant. Previous studies have shown that Arg580 mutants have a significantly weakened affinity for guanine nucleotides and are cytotoxic when expressed in cells. Here, we found that when the TG2 R580K mutant is exposed to sufficiently high concentrations of GTP, it forms an open-state dimer that binds GTP, possibly through interactions with both open-state monomers. This dimer is unable to be reduced to a closed-state monomer by the addition of more GTP, due to an inability for GTP binding to overcome the flexibility of the β-barrel domains without the stabilizing interaction with Arg580. Also, the TG R580K mutant in the presence of GTP transitions to higher order oligomeric species at lower Ca²⁺ concentrations than wildtype TG2, which may explain its cytotoxic effects. Furthermore, we found that the open state dimer also forms when wildtype TG2 is treated with Ca²⁺; however, the TG2 dimer that forms in the presence of Ca²⁺ can be reversed back to the closed state monomer in the presence of GTP. We propose that the TG2 oligomers stabilized by Ca²⁺ can be reversed by GTP through its interaction at the open-state dimer binding site, which favors the disassembly of the higher-order species as found in the R580K mutant. Importantly, due to the stabilizing interactions with Arg580 in wildtype TG2, GTP is able to bind to the closed state TG2 monomer by achieving a high-affinity interaction that can overcome the flexibility of the β-barrel domains. Interestingly, a C-terminal TG2 splice variant, termed "TG2-short", is expressed with a shortened β-barrel domain and has been shown to form cytotoxic 'aggregates' in cells independent of its nucleotide binding or crosslinking activity[74]. It is possible that with its weakened affinity for guanine nucleotides, TG2-short constitutively adopts a dimeric open conformation that more readily responds to intracellular levels of Ca²⁺ that then stabilize the higher-order oligomeric species responsible for its cytotoxic effects.

We have been interested in taking advantage of the cytotoxicity exhibited by open-state TG2 to develop small molecules capable of stabilizing this conformation, as a potential therapeutic strategy, including our lead compound, TTGM 5826[54]. However, our previous study did not identify how a molecule that does not bind to closed state TG2 could promote the formation of the open state in the presence of intracellular concentrations of GTP. Here, we report the structure of LM11, a molecule that is more potent in eliciting a cytotoxic effect in glioblastoma cells than its predecessor TTGM 5826, and investigate how it influences the conformational dynamics of TG2 using SAXS. We found that when TG2 is exposed to both Ca²⁺ and a small molecule open-state stabilizer such as LM11, it assumes conformational and oligomeric states that cannot be reversed by excess amounts of guanine nucleotides. In addition, our time-resolved SAXS experiments allowed us to observe the structural states of TG2 that accompany Ca²⁺ binding, in the presence and absence of LM11. We found that LM11 increases the rate at which Ca²⁺ binds and stabilizes the formation of higher-order oligomers as well as reduces the concentration of Ca²⁺ needed to overcome GTP-binding. Such open-state stabilizers may be especially effective in killing cancer cells given their potential to work together with low (physiological) concentrations of Ca²⁺ and overcome the relatively high concentrations of guanine nucleotides found in the cytoplasm of target cells.

Intriguingly, the discovery of the GTP-binding properties of the R580K dimer suggests an alternative route for drug design. For GTP to bind the open-state dimer, it would likely form interactions at the dimer interface, with the catalytic domain of one TG2 monomer (Phe174 and Lys173) and the β-barrel domains (Arg476 and Arg478) of a second TG2 monomer. However, given the reduced binding affinity of the R580K mutant for GTP, this suggests that the nucleotide binding site in the open-state dimer is not identical to that of the closed-state monomer indicating that it may be directly targeted by a small molecule inhibitor. Such an inhibitor might act as a staple to hold cytotoxic TG2 dimers together. We anticipate that future high-resolution structural studies seeking to characterize this conformation will lead to an improved understanding of TG2 biological and cellular signaling activities, and to an enhanced ability to

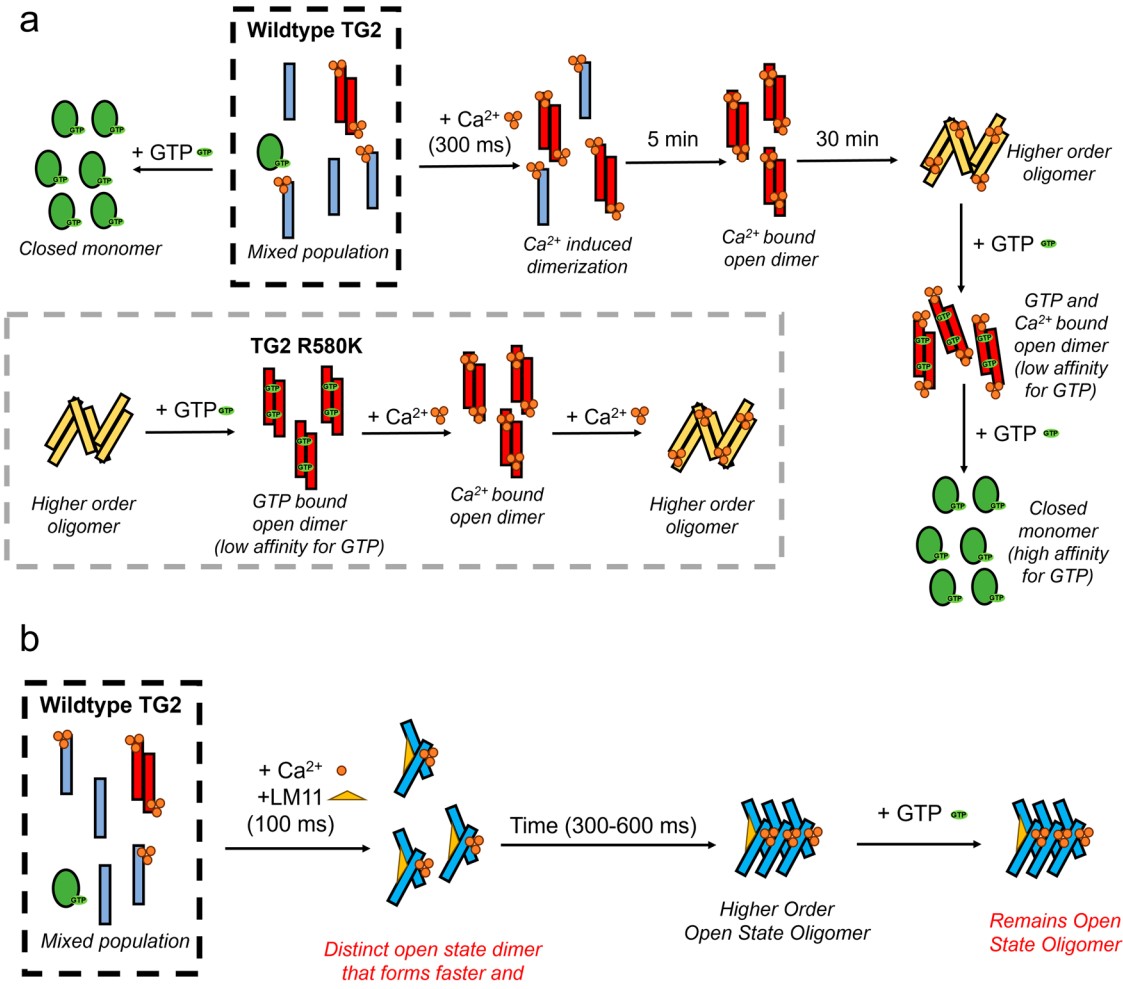

**Fig. 7 | Model for TG2 activation by Ca²⁺ in the presence and absence of the inhibitor, LM11. a** When GTP is added to TG2 in buffer, it can effectively adopt the closed conformation even in the presence of $Ca^{2+}$. In the absence of LM11, the addition of $Ca^{2+}$ to TG2 is accompanied by the formation of a dimer on short time scales (300 ms to 5 min) and higher order oligomers at longer time scales (30 min). This process is reversible by the addition of excess GTP, through interactions at the dimer interface. The R580K mutant exists as a higher order oligomer in the absence of calcium and GTP, but transitions to the nucleotide bound open state dimer upon addition of excess GTP. The addition of $Ca^{2+}$ to the R580K dimer is accompanied by the formation of higher order oligomers. **b** The presence of LM11 and $Ca^{2+}$ maintains TG2 in an open conformation which is not reversed by the addition of GTP. Moreover, TG2 adopts a dimeric species that has a distinct conformation to the dimer fraction observed in the absence of LM11. Even on short time scales less than one second, higher-order oligomers are observed which are irreversible by the addition of excess GTP.

design new small molecule inhibitors targeting the GTP-binding site of TG2 as well as more potent analogs of LM11 that will provide a therapeutic benefit to patients suffering from cancers exhibiting high TG2 expression.

## Methods
### Reagents
Common chemicals, BODIPY FL GTPγS, and other consumables were obtained from Thermo Fisher Scientific (Waltham, MA). DMEM, trypsin-EDTA, and fetal bovine serum (FBS) were purchased from Invitrogen (Waltham, MA). Competent cells were purchased from New England Biolabs (Ipswich, MA). TTGM 5826 was obtained from ChemBridge (San Diego, CA). Compounds LM1, LM2, LM3, and LM4 were from Specs (Narragansett, RI). All other compounds in the LM-series were from Vitas-M Laboratory (Champaign, IL). The PVDF transfer membrane and ECL reagent were purchased from Perkin-Elmer Life Sciences (Waltham, MA). The Vinculin (#13901) and anti-rabbit IgG, HRP-linked (#7074) antibodies were obtained from Cell Signaling Technology (Danvers, MA). The TG2 (MS-300-P) antibody was obtained from Neomarkers (Fremont, CA).

### TG expression and purification
Tissue transglutaminase (TG2) was expressed and purified as previously described[50]. Briefly, Escherichia coli BL21 (DE3) competent cells (New England Biolabs) were transformed with a pET28a vector containing the N-terminal six-histidine tagged human TG2. The transformed cells were grown in Luria broth (LB) with 50 μg/mL kanamycin at 37 °C until the OD600 reached 0.6–0.8, following which, protein expression was induced by treating the cells with 10 μM IPTG at 25 °C for 16–18 h. Next, the cells were lysed by sonication in lysis buffer (50 mM Tris pH 8.5, 500 mM NaCl, 0.1 mM PMSF, 5 mM β-mercaptoethanol (BME), 10% glycerol, 50 μM GTP) and the lysate was clarified by centrifugation at $185,000 \times g$ for 45 min. The supernatant was applied to a 5 mL HisTrap HP column (Cytiva Life Sciences) equilibrated with equilibration buffer (50 mM Tris pH 8.5, 500 mM NaCl, 10% glycerol, 0.1 mM PMSF, 5 mM BME), washed with 100 mL of equilibration buffer followed by 100 mL of wash buffer (50 mM Tris pH 8.5, 10 mM NaCl, 10% glycerol, 5 mM BME). The protein was eluted using wash buffer containing 320 mM imidazole. The eluent was then applied to a 5 mL HiTrap Q HP column (Cytiva Life Sciences) and eluted using a gradient of Buffer A (50 mM MES pH 6.5, 10 mM NaCl, 10% glycerol, 1 mM DTT) and Buffer B (Buffer A with

800 mM NaCl). The fractions containing the protein peak were pooled and applied to an additional 5 mL HisTrap column equilibrated with HisTrap Buffer A (20 mM HEPES pH 7.5, 100 mM NaCl, 10% glycerol), washed with 100 mL of HisTrap Buffer A, and then eluted over a gradient using HisTrap Buffer B (HisTrap Buffer A with 500 mM imidazole). The peak fraction was pooled and injected onto a HiLoad Superdex 200 (Cytiva Life Sciences) equilibrated with TG2 Storage Buffer (20 mM HEPES pH 7.5, 100 mM NaCl, 10% glycerol, 1 mM DTT) for size-exclusion chromatography purification. The purified TG2 was then concentrated to 1 mg/mL using a 10 kDa cutoff centrifugal filter (Jumbosep, PALL), flash-frozen in liquid nitrogen, and stored at −80 °C until further use.

## Cryo-EM sample preparation and data collection
The TG2 cryo-EM sample was prepared by incubating 0.3 mg/mL TG2 with 5 mM GDP or 2 mM CaCl$_2$ for 10 min on ice. Cryo-EM samples were frozen using a Vitrobot Mark IV (Thermo Fisher Scientific) maintained at 4 °C and 100% humidity. A 4.2 μL solution was applied to glow-discharged all-gold grids (UltrAuFoil, R1.2/1.3, 200 mesh) and the excess sample was blotted away for 2 s, then plunge-frozen into liquid ethane cooled with liquid nitrogen. CryoEM images were collected on a Talos Arctica (Thermo Fisher Scientific) with a Gatan GIF Quantum LS Imaging energy filter (20 kV slit), operated at 200 kV at a nominal magnification of 79,000× with a corresponding super-resolution pixel size of 0.517 Å and a 70 μm objective lens. Micrographs were recorded using a Gatan K3 direct electron detector with a dose rate of ~27.5 electrons/pixel/s for a total dose of 51.1 electrons/Å$^2$ and defocus values ranging from −1.2 μm to −2.4 μm. Images were collected using EPU (Thermo Fisher Scientific).

## Cryo-EM data processing, model building, and validation
All cryo-EM data processing was carried out using cryoSPARC version 4.2.1 (Structura Biotechnology)[75]. Detailed workflows are included in the Supplementary (Figs. S11 and S12). Briefly, Patch Motion Correction and Patch CTF estimation was used to prepare the micrographs. Micrographs were curated to include CTF fits better than 5 Å for downstream processing. Blob picking with a maximum radius of 140 Å was used to generate templates, which were then used for template picking. 2D classification was used to select particles that displayed clear secondary structure features and these particles were used for downstream processing. Selected particles were used in an ab initio reconstruction with two classes to investigate possible heterogeneity in the sample. Homogeneous refinement and nonuniform refinement were then used on to generate a consensus map[76]. Defocus refinement and local CTF refinement was carried out as implemented in cryoSPARC[77]. CryoEM maps were sharpened using Sharpening Tools in cryoSPARC. For TG2 bound to GDP, an initial atomic model was generated using ModelAngelo (version 1.0.1) using the sharpened cryoEM map and the protein sequence for human TG2 as defined in UniProt (TGM2: P21980) as input. The ModelAngelo output was then inspected and refined manually in WinCoot (Coot version 0.9.4.1). The model for TG2 bound to Ca$^{2+}$ was generated by using the crystal structure of TG3 bound to Ca$^{2+}$ and building a homology model using Prime (Schrödinger)[70,71]. The model was then visualized in ChimeraX[78].

## BODIPY-GTPγS binding assay
BODIPY FL GTPγS (500 nM final concentration) was added to the assay buffer (50 mM Tris pH 7.5 and 1 mM EDTA) in a 1 ml cuvette in the presence or absence of TTGM 5826 or LM11 (50 μM final concentration in assay). Once a stable background fluorescence measurement was achieved, purified TG2 recombinant protein (150 nM final concentration) was added, and fluorescent measurements were taken until the signal stabilized. Then, Ca$^{2+}$ was added in 10 mM increments. All measurements were made on a Varian Eclipse spectrofluorometer. The excitation and emission wavelengths were set at 504 nm and 520 nm, respectively.

## Cell culture and preparation of whole cell lysates
U87 and T98G glioblastoma cells (ATCC, Manassas, VA) were maintained at 37 °C, 5% CO$_2$, in DMEM containing 10% FBS. Cells were authenticated by the ATCC prior to purchase and were discarded for fresh stock after 3 months of use. Whole cell lysates were prepared by washing the cells with phosphate-buffered saline (PBS) and then lysing them with cell lysis buffer (25 mM Tris pH 7.5, 100 mM NaCl, 1% Triton X-100, 1 mM EDTA, 1 mM DTT, 1 mM Na$_3$Vo$_4$, 10 mM β-glycerol phosphate, and 1 μg/ml each of leupeptin and aprotinin).

## Cell death assay using trypan blue
U87 and T98G glioblastoma cell lines were grown in six-well plates to 50–70% confluency and washed 3 times with 2 mL of sterile PBS before being maintained in serum-free medium supplemented with the small molecules TTGM 5826 or LM11, or with the carrier DMSO, at the indicated concentrations. After 36 h, the attached and floating cells were collected, treated with a 1:1 ratio of 0.04% trypan blue (Gibco), and counted using a hemocytometer. The percentage of dead (blue) cells were plotted as a function of small molecule concentration.

## Western blots analysis
The protein concentration from U87 or T98G cell lysates was determined using Bradford Assay reagent (Bio-Rad). The lysates were normalized based on total protein concentration, resolved on a 4–20% gradient SDS-PAGE gel (Invitrogen), and the proteins were transferred to PVDF membrane (Thermo Fisher). The membrane was blocked in 5% (w/v) milk dissolved in TBST (19 mM Tris, 2.7 mM KCl, 137 mM NaCl, and 0.5% Tween-20), and then incubated overnight at 4 °C with either an antibody against Vinculin (Cell Signaling) or TG2 (Neomarkers) diluted 1:1000 in TBST. The next day, the membranes were incubated with an anti-rabbit or anti-mouse IgG HRP-linked antibody for one hour, washed with TBST, exposed to ECL (Perkin Elmer), and developed.

## Static SAXS
Initial screening of TG2 concentration, Ca$^{2+}$, GTP, and drug (TTGM 5826 or LM11) conditions were performed on a BioXolver (Xeoncs Inc., Holyoke, MA). All TG2 SAXS samples were prepared in TG2 Storage Buffer (20 mM HEPES pH 7.5, 100 mM NaCl, 10% glycerol, 1 mM DTT). We found that 2 mg/mL TG2, 2 mM CaCl$_2$, 5 mM GTP, and 50 μM inhibitor were optimal. Samples were incubated for 5 min with CaCl$_2$ and 30 min with inhibitor (TTGM 5826 or LM11). GTP was added right before the measurement. These conditions were also measured at the Cornell High Energy Synchrotron Source (CHESS) beamline ID7A1 using their standard setup (beam energy 11.3 keV, $q$ range 0.006–0.7 Å$^{-1}$) to get data with a higher signal to noise ratio. All data were normalized with the pin diode measurements on the beam stop. All data reported in this manuscript were collected on the ID7A1 beamline.

## Time-resolved SAXS
Time-resolved SAXS measurements were taken as described in our previous work (Zielinski et al.[66]). Briefly, a Kenics-style chaotic advection

**Table 2 | Flow conditions through the Kenics mixer for the different timepoints. Beam position reflects the distance from the tip of the Kenics mixer**

| Timepoint (ms ± uncertainty) | Sample A Flowrate (μL/min) | Sample B Flowrate (μL/min) | Sheath Flowrate (μL/min) | Beam Position (μm) |
|---|---|---|---|---|
| 32 ± 6 | 60 | 60 | 203.1 | 450 |
| 63 ± 6 | 60 | 60 | 203.1 | 691 |
| 100 ± 13 | 30 | 30 | 101.5 | 750 |
| 316 ± 18 | 20 | 20 | 67.7 | 1970 |
| 631 ± 23 | 20 | 20 | 67.7 | 3058 |

mixer was used to rapidly combine two species just prior to data collection. Different timepoints were reached by varying the flowrate through the mixer or the position of the X-ray beam relative to the tip of the mixer (Table 2). For all time-resolved measurements, the initial TG2 concentration was 2 mg/mL and the final, probed concentration was 1 mg/mL. The final concentrations for $Ca^{2+}$ and LM11 were 2 mM and 50 μM respectively. $CaCl_2$ and LM11 were added to the sheath at the final, mixed concentration to mitigate these small species from diffusing radially out of the sample stream. All data were normalized with a semi-transparent beam stop made of 250 μm thick molybdenum foil. For all timepoints, buffer and sample data collection were taken sequentially to ensure good matches for buffer subtraction.

## SAXS data analysis

All SAXS data, static and time-resolved, were analyzed with BioXTAS Raw software[79–81]. For each measurement, buffer matches (without protein, from the flowthrough of TG2 purification) were also collected. These profiles were then subtracted from the protein sample measurements to get the scattering from the protein alone. Guinier analysis was used to calculate the radius of gyration ($R_g$) and the forward scattering (I(0)). CRYSOL and OLIGOMER modeling was done using ATSAS version 3.2.1[82,83]. Pair-distance distribution analysis was carried out in RAW using GNOM and bead models were generated using DAMMIF[84,85]. ChimeraX was used to visualize SAXS envelopes. Further data processing, such as the deconvolution, was performed using in-house MATLAB scripts. More specifically, the in-house deconvolution scripts (provided in the supplementary materials) determined the minimum of the error squared by varying a and b in the following function, $I_{data} - aI_a - bI_b$, where $I_a$ and $I_b$ were input models and $I_{data}$ is the data being deconvolved. We used the calculated scattering profiles for the open state monomer and open state dimer presented in Fig. S2 as the inputs.

## Docking analysis

The LM series of compounds were identified by searching for commercially available chemicals with high structural similarity to TTGM 5826. The crystal structure of TG2 bound to a peptidomimetic inhibitor (PDB: 2Q3Z) and the LM series of compounds were prepared for docking using the Protein Preparation Wizard and LigPrep in Maestro (Schrödinger, Inc)[86]. The LM series was docked using Glide and manually analyzed in Maestro (Schrödinger, Inc)[72]. Pharmaceutically relevant properties of the LM series were determined using QikProp (Schrödinger, Inc).

## TG2 crosslinking assay

Recombinant TG2 (43 nM) was mixed with each LM series compound, or DMSO (as a control), for 5 min at room temperature in reaction buffer (10 mM Tris, 100 mM NaCl, pH 7.4). The crosslinking reaction was initiated by the addition of 10 mM DTT, 10 mM $CaCl_2$, 62.5 μM biotinylated pentylamine, and N,N-dimethyl casein. The reaction was resolved on a 4–20% gradient SDS-PAGE gel (Invitrogen), and the proteins were transferred to PVDF membrane (Thermo Fisher). The membrane was blocked in 10% bovine serum albumin (BSA) in BBST (20 mM sodium tetraborate, 100 mM boric acid, 80 mM sodium chloride, and 1.5% Tween-20, pH 8.5), incubated overnight at 4 °C, and probed with 1:2000 HRP-conjugated streptavidin and 5% BSA in BBST.

## Reporting summary

Further information on research design is available in the Nature Portfolio Reporting Summary linked to this article.

## Data availability

SAXS data has been deposited into the Small Angle Scattering Biological Data Bank (SASBDB: SASDTL3, SASDTM3, SASDTN3, SASDTP3, SASDTQ3, SASDTR3, SASDTS3, SASDTT3, SASDTU3, SASDTV3, SASDTW3, SASDTX3, SASDTY3, SASDTZ3, SASDT24, SASDT34, SASDT44, SASDT54, SASDT64, SASDT74, SASDT84, SASDT94,

SASDTA4, SASDTB4, SASDTC4, SASDTD4, SASDTE4, SASDTF4, SASDTG4, SASDTH4, SASDTJ4, SASDTK4, SASDTL4, SASDTM4, SASDTN4, SASDTP4, SASDTQ4, and SASDTR4). CryoEM data has been deposited to the Protein Data Bank (PDB ID: 8TR9) and the Electron Microscopy Database (EMDB IDs: 41574 and 42356). All other data are contained in the manuscript and accompanying supplemental information (see Supplementary Data).

## Abbreviations

| | |
|---|---|
| TG2 | Tissue Transglutaminase 2 |
| SAXS | Small angle X-ray scattering |
| TR-SAXS | Time-resolved small angle X-ray scattering |
| Rg | Radius of Gyration |
| GDP | Guanosine diphosphate |
| GTP | Guanosine triphosphate |
| $Ca^{2+}$ | Calcium |

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

## Acknowledgements

We would like to thank Dr. Lee McDermott for significant assistance in selecting the LM-series compounds. We would also like to thank Drs. Richard Gillian and Quinqiu Huang for their help setting up data collection at CHESS beamline ID7A1; Dr. Marc Antonyak for helpful discussions on TG2 biology; Robert C. Miller for advice on SAXS modeling and analysis; and undergraduate student Sydney Devore for assistance with fluorescence assays. CryoEM data collection was carried out at the Cornell Center for Materials Research (CCMR), and we thank Drs. Mariena Silvestry-Ramos and Katie Spoth, for assistance with cryoEM sample preparation and data collection (NSF, DMR-1719875). This work was supported by grants from the National Institutes of Health (NIH), National Institute of General Medical Sciences (NIGMS) (grant Nos. R35-GM122514 awarded to L.P.; T32-GM008267 awarded to C.A.; R35GM122175, CA201402 and CA223534 awarded to R.C.), and the National Science Foundation (NSF), Directorate for Biological Sciences (grant Nos. DBI-1930046; STC1231306 awarded to L.P.). SAXS data were acquired at the Sector 7A1 beamline at CHESS. CHEXS is supported by the NSF award DMR-1829070, and the MacCHESS resource is supported by NIGMS award 1-P30-GM124166-01A1 and NYSTAR. SAXS data were also acquired on a Xenocs BioXolver acquired through NIH (grant No. S10OD028617).

## Author contributions

K.Z., S.P., S.K.M., and C.A. performed SAXS sample preparation and data collection. C.A., K.Z., and S.P. analyzed SAXS data. C.A. performed SAXS modeling with DAMMIF, CRYSOL, and OLIGOMER and K.Z. performed TR-SAXS modeling with the SAXS deconvolution. K.Z. made the Kenics mixers and designed the microfluidic channels for the TR-SAXS experiments. C.A. performed cryoEM sample preparation, data collection, and analysis. S.K.M. and C.A. purified recombinant TG2. S.K.M. and D.O. performed the cell death assays and western blot experiments. S.K.M. and D.O. performed fluorescence BODIPY-GTP binding assays. W.P.K. performed the TG2 crosslinking activity assays. W.P.K. and C.A. performed docking analysis of TG2 inhibitors. C.A. wrote the manuscript, with feedback from all authors. R.A.C. and L.P. led the project and secured funding.

## Competing interests

The authors declare no competing interests.
