## [Peer Review File · Communications Biology]

Reviewers' comments:

Reviewer #1 (Remarks to the Author):

Aplin/Zielinski et al report on the conformational states of the enzyme transglutaminase 2 (TG2) that exists in the presence of different cellular factors (GTP, Ca²⁺) and how a novel small molecule can modulate this equilibrium, shifting the enzyme to a form that is cytotoxic in cancer cells. The authors marry cryo-electron microscopy with static and time-resolved SAXS measurements to dissect the ligand-dependent transitions in conformation and oligomeric state, overcoming the limitations of X-ray structural models obtained from a crystal lattice.

While SAXS had already been applied in the past on this enzyme (Kim et al 2017 Amino Acids, Singh et al JBC 2016, Casadio et al Eur J Biochem 1999), the implementation of time-resolved SAXS is novel to probe these transitions in response to effector is novel, as is its use to drive the main motivation of this study: structural insights into small molecules that can modulate the conformational state of the target to a form that is cytotoxic to cancer cells. This exciting concept makes this study a very appropriate fit for the broader audience of Communication Biology.

Overall, the study is well-written and thorough. This reviewer's requests for revision focus on the SAXS analyses and interpretation.

Major:

1. The continued reference to "higher-order oligomers" | The authors do not provide any complementary solution analyses that corroborate their interpretation of the species present in the SAXS data, which is complicated by the existence of not only different conformers, but different oligomers of the enzyme. Can they provide any such complementary analysis (e.g. SEC, SEC-MALS, AUC) that can confirm their understanding of the solution mixture in the presence of different effectors for their preparations of this enzyme, at the temperature and concentrations used in the SAXS measurements? For example, is it possible that there are also tetramers and other higher order oligomers are present in SAXS measurements that are interpreted with monomer and dimer models?
2. Concentration-dependence vs scattering signal | the authors point out that they optimized their SAXS signal to a specific concentration of enzyme. Representative concentration-dependent data should be provided in supplemental to show the impact of sample concentration on R_g and $I(0)$. Does SVD analysis (model-independent) support their assignment of the number of species?
3. Figure 2B | Of course, R_g can change due to oligomerization, conformational changes, or both. The authors should add $I(0)$ or experimental Mass as a double Y plot to help the reader track oligomeric state as they correlate to changes in R_g . Same for figure 2E and Figure 3F, Figure 5C.
4. Figure 3D – residuals between $\sim 0.1-0.15$ q seems discrepant in a number of comparisons like this. Does rigid body (CORAL, SASREF) modeling further improve the correlation in this region?

Those q ranges would correlate well with expected domain-domain distances. For these comparisons with structural models throughout the study, either P_r /shape distribution comparisons, or residuals for the comparison between I_{exp} and the I_{model} should be additionally provided as a subpanel.

Atomic inventory | the authors should spell out the atomic inventory of the models used in their structural analyses and how they reconcile with their full-length constructs used in experiments. A discrepancy in inventory could explain such slight differences in different local regions of the scattering profile.

Figure S1: Please provide the calculated R_g and D_{max} for each of these experimental models (eg HullRad, US-SOMO, Hydropro), as a point of reference, prior reported values for monomers and dimers.

5. Figure S3 – fonts are way too small on X and Y axes. Also, it appears that there is still significant discrepancy in the Guinier region between the Oligomer fits and the experimental data. Please show Guinier plots.

6. Interpretation of time-resolved data | Of course, the challenge in this analysis is distinguishing the components of the mixtures that exist. Yet, the authors provide very little information about their “In-house deconvolution” scripts. More experimental details and the scripts themselves should be provided. How does it differ from the OLIGOMER program? Is SVD performed?

Minor

7. Fix citation format for Datta et al in Methods, TG Expression and Purification.

8. Specifically report what buffer was employed for SAXS measurements (add to methods). Are the data in absolute or arbitrary units of intensity? Why are Figures 2A and 3A so different in magnitude on the y-axis? (prior data? If so, call out in figure legend)

9. The accessible q should be reported in the methods, along with at least some details of the experimental data collection conditions at CHESS.

10. Table S1 – please reformat to provide more clarity on the composition of each SAXS profile recorded (it is very confusing to follow). Is $I(0)$ in arbitrary or absolute units?

11. Table S4 and S5 – provide expected Molecular Weight for monomers and dimers in the table legend for the reader’s reference.

12. Figure 1 – To help a broader audience better appreciate, can you place a box around the area on the models in Figures A-B that corresponds to drug binding site depicted in Figures C-D?

Reviewer #2 (Remarks to the Author):

In their manuscript, the authors report sophisticated conformational studies of TG2, a protein known to play different biological roles depending on its conformational state. They bring to bear cutting edge methodology (SAXS and cryo-EM) focussed on important questions regarding the conformationally-dictated roles of TG2 in cancer cells, with important implications on the design of therapeutically relevant drugs. They provide solid evidence for the effect of various ligands on conformation of TG2, as well as very important information on the effect of the authors' new inhibitors on conformation (and assumed activity), in the context of cancer treatment.

This manuscript offers many important results.

This reviewer thinks the cryo-EM results showing GTP-bound TG2 as a monomeric species is a very important contribution, beyond Cerione's pioneering contribution of the Xray structure (showing a crystallographic dimer).

The proposal that the conservative R580K mutant TG2 may form an open state dimer that can bind GTP is interesting. It is not yet clear if this is physiologically relevant, or an event that occurs in an in vitro experiment, but this is an important observation to report.

I think this is very important work and it should be published, however I do have some concerns.

Most importantly, for an article that focusses on the conformational states of TG2, this reviewer believes the language regarding conformational changes should be more rigorous. For example, if we consider the two conformations of TG2, the 'open' published by Khosla and the 'closed' published by Cerione himself, we can see that the GTP binding site of the closed conformation is completely disrupted in the open conformation (see Figure 1, Eur J Med Chem 2022, 232, 11417). The GTP binding site does not exist in the open conformation. So it is not clear to this reviewer why the authors would refer to GTP binding as inducing a conformational change (as on Page 3: "upon binding GDP or GTP, TG2 adopts a "closed" conformation"). Where would GTP bind to the open form, to induce it to close? It does not seem to be appropriate to invoke an induced fit mechanism in the case of such a profound conformational change. In an induced fit mechanism, a binding site that is partially complementary to the ligand initially undergoes a relatively minor conformational change to become more complementary to the ligand. This is not possible with TG2. There is no GTP binding site in the open form of TG2, where GTP could bind to induce the enzyme to literally fold in half. I think it is infinitely more likely that the closed conformation is first adopted, forming the GTP binding site that then binds GTP or GDP, further stabilizing that conformation. The sentence on page 3 could be changed to "... have revealed that the "closed" conformation can bind GDP or GTP, stabilizing a conformation that occludes the..."

On page 6, the authors write that addition of GTP "enabled wildtype TG2 to adopt a compact conformational state". However, this experiment does not reflect how a conformational state is adopted, it merely reports which one is more favored. More strictly speaking, "increasing additions of GTP increased the concentration of a compact conformational state for wildtype TG2".

Similarly, the authors write "Upon Ca²⁺ binding, TG2 is thought to undergo a conformational change..." and on page 6, "...the ability of calcium to induce TG2 to assume an open conformational

state...” I acknowledge that this loose language has been published and re-published in the TG2 literature, but do the authors really believe that binding a calcium ion can induce such a dramatic conformational change? I suggest that it would be more accurate to state that the open conformation of TG2 is stabilized by calcium binding, without claiming that calcium binding induces the conformational change. That is, the conformational change probably happens first, and then calcium binding shifts the equilibrium in favor of the open form.

On page 8, the authors report that calcium “promotes the dissociation of guanine nucleotides from TG2 as an outcome of a conformational change”. Do the cited references 4 and 42 provide evidence for the promotion of a conformational change? Or rather, more strictly speaking, that in the presence of calcium, the conformational equilibrium is shifted to favor a certain conformation that does not bind guanine nucleotides?

I understand that my comments could be reduced to a debate over “induced fit” versus “conformational capture” binding mechanisms, but I believe the authors have the opportunity and responsibility to be very careful with the language in their manuscript. Currently, the language they are using is uniquely consistent with the induced fit model. Now, I believe the “conformational capture” argument is better aligned with other points the authors make in their article. (For example, even inside the cell, where calcium concentrations are low, TG2 is probably able to sample the open conformation, at least fleetingly, and we know this because TG2 is capable of exhibiting some cross-linking activity intracellularly. And describing their inhibitor TTGM 5826 as being able to “trap TG2 in an open state” sounds perfect to me.) But I will not ask the authors to adopt the “conformational capture” bias, either. All I am requesting is that the authors simply remove the mechanistic bias from their text. They have excellent evidence for how various ligands and inhibitors affect the conformational equilibria of wildtype and mutant TG2. That is, they have excellent evidence on the effect on the equilibria end points. They do not have any evidence for an induced fit mechanism of conformational modulation. So instead of suggesting that calcium “promotes the dissociation of GTP”, they could simply state that they have shown clearly that in the presence of calcium, less GTP is bound by TG2. Instead of referring to calcium levels that are “required to drive TG2 to conformational transition”, what they are really observing are “calcium levels required to displace the conformational equilibrium of TG2”. Their language should not comment on the transition, but rather on the endpoint.

This language is also relevant to their mechanism of inhibition. On page 16, the authors imply that a molecule that does not bind to the closed state of TG2 could promote the formation of the open state, in the presence of intracellular concentrations of GTP. But this is not at all difficult to explain by the conformational capture mechanism: TG2 is always in dynamic equilibrium between its open and closed forms, either inside the cell or out of it. GTP binds the closed form and stabilises it, pulling the equilibrium more towards the closed side, but there will always be a fraction of TG2 that releases GTP and samples the open conformation... When TTGM 5826 or LM11 are added to the cell, they bind tightly to the open conformation, when it is formed, and by stabilizing it, they shift the conformational equilibrium back towards the side of the open conformation. All these binding events are reversible, so the final endpoint will simply be determined by the relative affinities of closed form TG2 for GTP and open form TG2 for inhibitor. So rather than combining the conformational change and calcium/GTP binding in one equilibrium (as in Figure 1 of this manuscript) the equilibria are probably separate as in: $TG2_{cl}.GTP \leftrightarrow TG2_{cl} \leftrightarrow TG2_{op} \leftrightarrow$

TG2op.LM11 (as in Figure 1 of Eur J Med Chem 2022, 232, 11417).

If LM11 is able to capture the fleeting, open conformation of TG2 in a cell better than other inhibitors, it is probably simply because it has higher affinity for the open conformation than other inhibitors, and is bound more tightly. It does not necessarily mean LM11 is able to intervene in the interaction with GTP, or in the physical transition from the closed form to the open form.

Another aspect of the writing of this manuscript that requires attention is the context of 'cell permeability of inhibitors' that the authors intend to establish. First of all, the authors imply that 'peptidomimetic inhibitors' and 'peptides' are the same thing (page 4). They are not.

Peptidomimetic inhibitors are not peptides, like the inhibitor shown in Figure 1D. They are small molecule inhibitors that contain amide linkages that resemble the peptide substrates of TG2. They do not all suffer from the same drawbacks listed on page 4 as peptides do, such as the one shown in Figure 1D.

Secondly, the authors claim that described TG2 inhibitors exhibit poor cell permeability, in support of which they cite their review article from 2018. However, many peptidomimetic small molecule inhibitors have been SHOWN to exhibit good cell permeability, in explicit permeability assays (e.g. J. Med. Chem. 2012, 55, 1021; ACS Med. Chem. Lett. 2012, 3, 1024; Chem. Biol. 2015, 22, 1347; Eur J Med Chem 2022, 232, 114172; FEBS J. 2023, 290, 5411).

Third, they imply that their small molecule inhibitors are superior to previous inhibitors with respect to permeability. However, to the best of my knowledge, the authors have never actually tested or reported the cell permeability of their inhibitors (either TTGM 5826 (Ref 54) or LM11) to make a legitimate quantitative comparison. In this manuscript they only report values predicted by QikProp.

So overall, the authors have 1) made implications about past inhibitors that are not accurate, and 2) made implications about their own inhibitors that are founded on calculated predictions, not on facts. This appears to be somewhat negligent writing, but it is easily corrected. The authors should send a sample of TTGM 5826 and LM 11 to their favourite CRO, and have them run a permeability assay (for example, bidirectional MDCK if they want to compare to their QikProp predictions and to published literature values). Then they will have the numbers to compare their inhibitors to those already evaluated and reported in the literature. (I strongly suspect the authors' inhibitors WILL be superior in this regard, but claims of superiority should be based on facts, not predictions.)

We sincerely thank the reviewers for their insightful comments and general support of our study. We have revised the manuscript to address their concerns. Our point-by-point response to each of the reviewers' comments is provided below.

Reviewer #1 (Remarks to the Author):

Aplin/Zielinski et al report on the conformational states of the enzyme transglutaminase 2 (TG2) that exists in the presence of different cellular factors (GTP, Ca²⁺) and how a novel small molecule can modulate this equilibrium, shifting the enzyme to a form that is cytotoxic in cancer cells. The authors marry cryo-electron microscopy with static and time-resolved SAXS measurements to dissect the ligand-dependent transitions in conformation and oligomeric state, overcoming the limitations of X-ray structural models obtained from a crystal lattice.

While SAXS had already been applied in the past on this enzyme (Kim et al 2017 Amino Acids, Singh et al JBC 2016, Casadio et al Eur J Biochem 1999), the implementation of time-resolved SAXS is novel to probe these transitions in response to effector is novel, as is its use to drive the main motivation of this study: structural insights into small molecules that can modulate the conformational state of the target to a form that is cytotoxic to cancer cells. This exciting concept makes this study a very appropriate fit for the broader audience of Communication Biology.

Overall, the study is well-written and thorough. This reviewer's requests for revision focus on the SAXS analyses and interpretation.

Major:

1. The continued reference to "higher-order oligomers" | The authors do not provide any complementary solution analyses that corroborate their interpretation of the species present in the SAXS data, which is complicated by the existence of not only different conformers, but different oligomers of the enzyme. Can they provide any such complementary analysis (e.g. SEC, SEC-MALS, AUC) that can confirm their understanding of the solution mixture in the presence of different effectors for their preparations of this enzyme, at the temperature and concentrations used in the SAXS measurements? For example, is it possible that there are also tetramers and other higher order oligomers are present in SAXS measurements that are interpreted with monomer and dimer models?

The reviewer makes an excellent suggestion to utilize SEC-MALS to assess the oligomeric state of TG2. We performed SEC-MALS for TG2 at 2 mg/mL in storage buffer and TG2 at 2 mg/mL + 2 mM Ca²⁺. For TG2 in storage buffer, we get a single broad peak corresponding to a molecular weight of 95.02 ± 1.33 kDa, which is just slightly lower than the molecular weight determined by SAXS (115.6 kDa, more details below). This difference is likely due to the dilution of TG2 as it goes through the SEC column. However, it demonstrates that without Ca²⁺, there is not a higher order oligomer fraction. For TG2 + Ca²⁺, we get a broad peak with a clear shoulder, indicating the presence of higher order oligomers. The molecular weight increases to 332.90 ± 8.74 kDa.

Based on our above results, we think it is reasonable that there are only appreciable higher order oligomer fractions when TG2 is incubated with Ca²⁺ for long periods of time. Otherwise, the sample seems to primarily be in the monomer and dimer states. This is further explained in point 2, with our SVD results, and in point 6, with more details about how the time-resolved deconvolutions were performed.

We have added a new Figure S5 to the supplementary materials, which shows the TG2 concentration series and the SEC-MALS results. To the Results section: *Characterization of TG2 conformational states stabilized upon the binding of Ca²⁺*, we added the following sentence (in bold): “Our experiments also suggest that the ability of TG2 to dimerize is dependent on protein concentration. The SAXS experiments were carried out at 25 μ M TG2 to achieve a sufficient scattering signal, while the cryo-EM analysis was carried out using 3 μ M TG2 as was necessary to obtain an appropriate particle density, which made it possible to capture the monomeric form of the TG2 open state. **A SAXS concentration series and SEC-MALS experiment for TG2 in storage buffer were performed to further demonstrate the concentration dependence on dimerization (Figure S5).**”

2. Concentration-dependence vs scattering signal | the authors point out that they optimized their SAXS signal to a specific concentration of enzyme. Representative concentration-dependent data should be provided in supplemental to show the impact of sample concentration on R_g and I(0). Does SVD analysis (model-independent) support their assignment of the number of species?

We have a concentration series at 1 mg/mL, 2 mg/mL, and 5 mg/mL of TG2 in the carrier buffer with no additives. At 5 mg/mL, there is a slight upturn at low q, the R_g value is 46.1 +/- 0.33 Å, the I(0) value is 0.13, the molecular weight by V_p is 134.8 kDa. For 1 mg/mL and 2 mg/mL, the I(0) was 0.11, the R_g was 40.3 +/- 0.50 Å vs 41.4 +/- 0.34 Å, and the molecular weight by V_p was 109.7 kDa vs 115.6 kDa, respectively. This discrepancy seems to be due to concentration dependent aggregation, and thus, 2 mg/mL was determined to be the optimal concentration.

We performed SVD on the concentration series (1 mg/mL, 2 mg/mL, and 5 mg/mL) and got 2 major states and one minor state. The coefficients for the components are -3.759 (with a corresponding negative singular vector), 11.068, and 0.006, demonstrating that the vast majority of our sample is in only two distinct states. Additionally, a linear combination of the first two components (with the appropriate coefficients) are an excellent match to the experimental data.

We have added the concentration series scattering profiles as a new Figure S5.

3. Figure 2B | Of course, R_g can change due to oligomerization, conformational changes, or both. The authors should add I(0) or experimental Mass as a double Y plot to help the reader track oligomeric state as they correlate to changes in R_g. Same for figure 2E and Figure 3F, Figure 5C.

We thank the reviewer for this helpful comment. All of the appropriate figures have been updated to include a double Y-axis with the Molecular Weight estimate determined from the Porod Volume.

4. Figure 3D – residuals between ~0.1-0.15 q seems discrepant in a number of comparisons like this. Does rigid body (CORAL, SASREF) modeling further improve the correlation in this region? Those q ranges would correlate well with expected domain-domain distances. For these comparisons with structural models throughout the study, either Pr/shape distribution comparisons, or residuals for the comparison between I_{exp} and the I_{model} should be additionally provided as a subpanel.

Atomic inventory | the authors should spell out the atomic inventory of the models used in their structural analyses and how they reconcile with their full-length constructs used in

experiments. A discrepancy in inventory could explain such slight differences in different local regions of the scattering profile.

We have added a new Figure S1 which includes a comparison of P(r) distributions. In addition, our atomic models used for CRYSOLO are based off of high-resolution structural models that are missing some amino acid residues. As requested, this information has been added to Figure S1 (now Figure S2) to make it clear what is included in each of the models. All of our constructs also have 6xHis tags which are not included in the atomic model. Taken together, these could explain discrepancies between the structural models and our experimental scattering profiles.

Figure S1: Please provide the calculated R_g and D_{max} for each of these experimental models (eg HullRad, US-SOMO, Hydropro), as a point of reference, prior reported values for monomers and dimers.

The R_g and D_{max} were calculated from theoretical scattering profiles determined using the FoXS server. These values are now included in the updated Figure S1 (now Figure S2).

5. Figure S3 – fonts are way too small on X and Y axes. Also, it appears that there is still significant discrepancy in the Guinier region between the Oligomer fits and the experimental data. Please show Guinier plots.

Figure S3 (Now Figure S4) has been updated to include a larger font size on the X- and Y- axis. We also added Guinier plots of the Oligomer fits. For TG2 in storage buffer, Figure S3A (now Figure S4A) shows that the Oligomer fits that include a dimer species are a good fit for the experimental data in the Guinier region while the Oligomer fit containing just monomeric TG2 species is a poor fit. For TG2 in calcium, Figure S3B (now Figure S4B) shows that both Oligomer fits consisting of >95% open state dimer is a good fit for the experimental data in the Guinier region.

6. Interpretation of time-resolved data | Of course, the challenge in this analysis is distinguishing the components of the mixtures that exist. Yet, the authors provide very little information about their “In-house deconvolution” scripts. More experimental details and the scripts themselves should be provided. How does it differ from the OLIGOMER program? Is SVD performed?

Our in-house deconvolution scripts determine the weights (a and b) of the following function, $I_{data} = aI_a + bI_b$, where I_a and I_b were input models and I_{data} is the data being deconvolved. This is achieved by finding the minimum of the error squared via the `fminsearch` function in MATLAB. We used the calculated scattering profiles for the open state monomer and open state dimer presented in Figure S1 as the inputs. Using these models as inputs aided us in making interpretation of the time-resolved data more physical. The code has now been added to the supplementary materials.

We did use OLIGOMER to perform a similar analysis, but the chi-squared values from our scripts were lower. SVD is not performed within our deconvolution scripts. We did also try three part fit, with the closed monomer as the third state, but the two state fit had lower errors.

The following has been added to the SAXS Data Analysis Methods section: “More specifically, the in-house deconvolution scripts (provided in the supplementary materials) determined the minimum of the error squared by varying a and b in the following function, $I_{data} - aI_a - bI_b$, where I_a and I_b were input models and I_{data} is the data being deconvolved. We used the calculated scattering profiles for the open state monomer and open state dimer presented in Figure S2 as the inputs.”

Minor

7. Fix citation format for Datta et al in Methods, TG Expression and Purification.

This has been corrected in the revised manuscript.

8. Specifically report what buffer was employed for SAXS measurements (add to methods). Are the data in absolute or arbitrary units of intensity? Why are Figures 2A and 3A so different in magnitude on the y-axis? (prior data? If so, call out in figure legend)

The SAXS samples in this manuscript were prepared in TG2 Storage Buffer. The following has been added to the Methods section in the revised manuscript:

All TG2 SAXS samples were prepared in TG2 Storage Buffer (20 mM HEPES pH 7.5, 100 mM NaCl, 10% glycerol, 1 mM DTT).

The data are in arbitrary units of intensity. The y-axis in 3A has a small note that the values are scaled by “1e-5”, but this was in font that was too small. This has been corrected so that 2A and 3A are shown at the same magnitude on the Y-axis and show that there is no major difference in intensity.

9. The accessible q should be reported in the methods, along with at least some details of the experimental data collection conditions at CHESS.

The following edits (bolded) were made to the Methods: Static SAXS section of the text to include more information about the experimental conditions at CHESS: “These conditions were also measured at the Cornell High Energy Synchrotron Source (CHESS) beamline ID7A1 using their standard setup (beam energy 11.3 keV, q range 0.006 - 0.7 Å⁻¹) to get data with a higher signal to noise ratio.”

10. Table S1 – please reformat to provide more clarity on the composition of each SAXS profile recorded (it is very confusing to follow). Is I(0) in arbitrary or absolute units?

Table S1 has been reformatted to improve clarity of the composition of each SAXS sample. I(0) is in arbitrary units. This has been included in the column header.

11. Table S4 and S5 – provide expected Molecular Weight for monomers and dimers in the table legend for the reader’s reference.

Tables S4 and S5 have been updated to include the expected molecular weight of the TG2 monomer (77 kDa) and the TG2 dimer (154 kDa).

12. Figure 1 – To help a broader audience better appreciate, can you place a box around the area on the models in Figures A-B that corresponds to drug binding site depicted in Figures C-D?

Figure 1 Panels A and B have been updated to include a red box outlining the drug binding site, as shown in Panels C and D.

Reviewer 2:

In their manuscript, the authors report sophisticated conformational studies of TG2, a protein known to play different biological roles depending on its conformational state. They bring to bear cutting edge methodology (SAXS and cryo-EM) focussed on important questions regarding the conformationally-dictated roles of TG2 in cancer cells, with important implications on the design of therapeutically relevant drugs. They provide solid evidence for the effect of various ligands on conformation of TG2, as well as very important information on the effect of the authors' new inhibitors on conformation (and assumed activity), in the context of cancer treatment.

This manuscript offers many important results.

This reviewer thinks the cryo-EM results showing GTP-bound TG2 as a monomeric species is a very important contribution, beyond Cerione's pioneering contribution of the Xray structure (showing a crystallographic dimer).

The proposal that the conservative R580K mutant TG2 may form an open state dimer that can bind GTP is interesting. It is not yet clear if this is physiologically relevant, or an event that occurs in an in vitro experiment, but this is an important observation to report. I think this is very important work and it should be published, however I do have some concerns.

Most importantly, for an article that focusses on the conformational states of TG2, this reviewer believes the language regarding conformational changes should be more rigorous. For example, if we consider the two conformations of TG2, the 'open' published by Khosla and the 'closed' published by Cerione himself, we can see that the GTP binding site of the closed conformation is completely disrupted in the open conformation (see Figure 1, Eur J Med Chem 2022, 232, 11417). The GTP binding site does not exist in the open conformation. So it is not clear to this reviewer why the authors would refer to GTP binding as inducing a conformational change (as on Page 3: "upon binding GDP or GTP, TG2 adopts a "closed" conformation"). Where would GTP bind to the open form, to induce it to close? It does not seem to be appropriate to invoke an induced fit mechanism in the case of such a profound conformational change. In an induced fit mechanism, a binding site that is partially complementary to the ligand initially undergoes a relatively minor conformational change to become more complementary to the ligand. This is not possible with TG2. There is no GTP binding site in the open form of TG2, where GTP could bind to induce the enzyme to literally fold in half. I think it is infinitely more likely that the closed conformation is first adopted, forming the GTP binding site that then binds GTP or GDP, further stabilizing that conformation. *The sentence on page 3 could be changed to "... have revealed that the "closed" conformation can bind GDP or GTP, stabilizing a conformation that occludes the..."*

We accept the reviewer's point here and have now revised the sentence in question to read "X-ray crystallography structures of TG2 have revealed that GDP or GTP binds to a "closed" conformational state that occludes the protein-crosslinking (transamidation) active site and thus eliminates its crosslinking activity (Fig. 1A) (37-40)."

In addition to addressing the other points raised by the reviewer, as described below, we have gone through the manuscript and made several corrections such that we no longer suggest that guanine nucleotides or Ca^{2+} 'induce' conformational changes within TG2 (or induce closed and open states, respectively). Rather, we agree that the better way to describe these interactions is that the closed and open states are in an equilibrium such that guanine nucleotides bind to and stabilize the closed state, while Ca^{2+} binds to and stabilizes the open state.

On page 6, the authors write that addition of GTP "enabled wildtype TG2 to adopt a compact conformational state". However, this experiment does not reflect how a conformational state is adopted, it merely reports which one is more favored. More strictly speaking, "increasing additions of GTP increased the concentration of a compact conformational state for wildtype TG2".

Based on the reviewer's comment, we have tried to be more precise in describing the effects of increasing GTP concentrations on TG2 structure. As suggested by the reviewer, the sentence in the revised manuscript now reads "increasing additions of GTP increased the amount of wildtype TG2 in a compact conformational state...".

Similarly, the authors write "Upon Ca^{2+} binding, TG2 is thought to undergo a conformational change..." and on page 6, "...the ability of calcium to induce TG2 to assume an open conformational state..." I acknowledge that this loose language has been published and re-published in the TG2 literature, but do the authors really believe that binding a calcium ion can induce such a dramatic conformational change? *I suggest that it would be more accurate to state that the open conformation of TG2 is stabilized by calcium binding, without claiming that calcium binding induces the conformational change. That is, the conformational change probably happens first, and then calcium binding shifts the equilibrium in favor of the open form.*

Here again, we concede the reviewer's point. On page 3, we have revised the sentence to read " Ca^{2+} binding stabilizes the open conformational state of TG2, which disrupts the nucleotide binding pocket as observed in X-ray crystal structures of TG2 bound to peptidomimetic inhibitors (Fig. 1B) (33, 42)." Likewise, we have revised the sentence referred to by the reviewer on page 6 to now read "...We show that its mode of action involves enhancing the ability of calcium to stabilize an open conformational state..."

On page 8, the authors report that calcium "promotes the dissociation of guanine nucleotides from TG2 as an outcome of a conformational change". Do the cited references 4 and 42 provide evidence for the promotion of a conformational change? Or rather, more strictly speaking, that in the presence of calcium, the conformational equilibrium is shifted to favor a certain conformation that does not bind guanine nucleotides?

We agree that perhaps a better way to describe the effects of Ca^{2+} on guanine nucleotide binding is in the following sentence now in the revised manuscript "It has been previously shown that Ca^{2+} stabilizes an open conformational state for TG2 that is no longer capable of binding guanine nucleotides (4, 42)."

I understand that my comments could be reduced to a debate over “induced fit” versus “conformational capture” binding mechanisms, but I believe the authors have the opportunity and responsibility to be very careful with the language in their manuscript. Currently, the language they are using is uniquely consistent with the induced fit model. Now, I believe the “conformational capture” argument is better aligned with other points the authors make in their article. (For example, even inside the cell, where calcium concentrations are low, TG2 is probably able to sample the open conformation, at least fleetingly, and we know this because TG2 is capable of exhibiting some cross-linking activity intracellularly. And describing their inhibitor TTGM 5826 as being able to “trap TG2 in an open state” sounds perfect to me.) But I will not ask the authors to adopt the “conformational capture” bias, either. All I am requesting is that the authors simply remove the mechanistic bias from their text. They have excellent evidence for how various ligands and inhibitors affect the conformational equilibria of wildtype and mutant TG2. That is, they have excellent evidence on the effect on the equilibria end points. They do not have any evidence for an induced fit mechanism of conformational modulation. *So instead of suggesting that calcium “promotes the dissociation of GTP”, they could simply state that they have shown clearly that in the presence of calcium, less GTP is bound by TG2. Instead of referring to calcium levels that are “required to drive TG2 to conformational transition”, what they are really observing are “calcium levels required to displace the conformational equilibrium of TG2”. Their language should not comment on the transition, but rather on the endpoint.*

We have now tried to do as the reviewer suggests above, as we do agree that strictly speaking, TG2 is ‘sampling’ closed and open conformational states and that guanine nucleotides bind to and stabilize the former while Ca^{2+} interacts with and stabilizes the latter.

This language is also relevant to their mechanism of inhibition. On page 16, the authors imply that a molecule that does not bind to the closed state of TG2 could promote the formation of the open state, in the presence of intracellular concentrations of GTP. But this is not at all difficult to explain by the conformational capture mechanism: TG2 is always in dynamic equilibrium between its open and closed forms, either inside the cell or out of it. GTP binds the closed form and stabilises it, pulling the equilibrium more towards the closed side, but there will always be a fraction of TG2 that releases GTP and samples the open conformation... When TTGM 5826 or LM11 are added to the cell, they bind tightly to the open conformation, when it is formed, and by stabilizing it, they shift the conformational equilibrium back towards the side of the open conformation. All these binding events are reversible, so the final endpoint will simply be determined by the relative affinities of closed form TG2 for GTP and open form TG2 for inhibitor. So rather than combining the conformational change and calcium/GTP binding in one equilibrium (as in Figure 1 of this manuscript) the equilibria are probably separate as in: $\text{TG2cl.GTP} \leftrightarrow \text{TG2cl} \leftrightarrow \text{TG2op} \leftrightarrow \text{TG2op.LM11}$ (as in Figure 1 of Eur J Med Chem 2022, 232, 11417). If LM11 is able to capture the fleeting, open conformation of TG2 in a cell better than other inhibitors, it is probably simply because it has higher affinity for the open conformation than other inhibitors, and is bound more tightly. It does not necessarily mean LM11 is able to intervene in the interaction with GTP, or in the physical transition from the closed form to the open form.

We tend to agree with the reviewer’s argument here and so we now state throughout the manuscript that Ca^{2+} and LM11 upon binding to the open form stabilizes this conformational state, which in turn would significantly weaken the affinity for guanine nucleotides.

Another aspect of the writing of this manuscript that requires attention is the context of 'cell permeability of inhibitors' that the authors intend to establish. *First of all, the authors imply that 'peptidomimetic inhibitors' and 'peptides' are the same thing (page 4). They are not. Peptidomimetic inhibitors are not peptides, like the inhibitor shown in Figure 1D. They are small molecule inhibitors that contain amide linkages that resemble the peptide substrates of TG2. They do not all suffer from the same drawbacks listed on page 4 as peptides do, such as the one shown in Figure 1D.*

We regret that this was the implication as we certainly recognize that peptidomimetic inhibitors and peptides are not the same. Thus, we have revised the sentences involved to eliminate any chance of this point being misconstrued.

Secondly, the authors claim that described TG2 inhibitors exhibit poor cell permeability, in support of which they cite their review article from 2018. However, many peptidomimetic small molecule inhibitors have been SHOWN to exhibit good cell permeability, in explicit permeability assays (e.g. J. Med. Chem. 2012, 55, 1021; ACS Med. Chem. Lett. 2012, 3, 1024; Chem. Biol. 2015, 22, 1347; Eur J Med Chem 2022, 232, 114172; FEBS J. 2023, 290, 5411).

We do appreciate that many peptidomimetic inhibitors have shown to be cell permeable and so we are now careful not to imply that most of these inhibitors are unable to enter cells.

Third, they imply that their small molecule inhibitors are superior to previous inhibitors with respect to permeability. However, to the best of my knowledge, the authors have never actually tested or reported the cell permeability of their inhibitors (either TTGM 5826 (Ref 54) or LM11) to make a legitimate quantitative comparison. In this manuscript they only report values predicted by QikProp.

So overall, the authors have 1) made implications about past inhibitors that are not accurate, and 2) made implications about their own inhibitors that are founded on calculated predictions, not on facts. This appears to be somewhat negligent writing, but it is easily corrected. *The authors should send a sample of TTGM 5826 and LM 11 to their favourite CRO, and have them run a permeability assay (for example, bidirectional MDCK if they want to compare to their QikProp predictions and to published literature values). Then they will have the numbers to compare their inhibitors to those already evaluated and reported in the literature. (I strongly suspect the authors' inhibitors WILL be superior in this regard, but claims of superiority should be based on facts, not predictions.)*

While we do not have a CRO, the reviewer raises a fair point that we lack direct evidence that the LM11 inhibitor is more permeable than various other small molecule inhibitors and so we no longer suggest that this is the case in the revised manuscript.

REVIEWERS' COMMENTS:

Reviewer #2 (Remarks to the Author):

As Reviewer #2 of the original manuscript, I remain convinced that this is important work that should be published.

I am relieved to read that the authors are in agreement with my interpretation of ligand binding vs conformational changes for TG2. I hope my original comments were not too pedantic, but I do think it is important to be precise. By tightening the language with respect to how ligand binding relates to conformational changes, the authors' manuscript will make a very positive contribution to the literature. Hopefully, others in the field will gain a more nuanced understanding of the timing of conformational changes and ligand binding, thanks to the authors' careful description and sound interpretation of their distinctive data.

The authors also modified their manuscript to describe the mechanism of action of LM11 (with respect to conformational modulation) more precisely.

The original manuscript made some unsupported statements regarding the cellular permeability of different classes of inhibitors. This was apparently corrected by removing all such commentary. In fact, the modified manuscript presents very little medicinal chemistry at all, and this Reviewer wonders whether any 'pharmaceutically relevant properties' of the LM series of inhibitors were even cited in the manuscript, even though this phrase remains in the experimental section on page 23. This reviewer looks forward to future work from the authors, describing their medicinal chemistry in more detail, while acknowledging that removing that discussion does not detract at all from the focus of the current manuscript.

In summary, the authors have accounted for all of my suggested corrections, and I do believe the manuscript is even stronger now. I congratulate the authors for their fine work, and look forward to following its impact on the field.

On a final note, on looking through the references (just now) I noticed many duplications. (Sorry, I should have caught this in the initial review.) For example, reference 68 appears to be the same as 37. Reference 69 is identical to 50. 74 is the same as 54. There may be others I haven't noticed, but this needs to be cleaned up.

We again thank the reviewers for their helpful comments and support of our study. We have revised the manuscript to address any remaining concerns. Our point-by-point response to each of the reviewers' comments is provided below.

Reviewer #1 (Remarks to the Author):

As Reviewer #2 of the original manuscript, I remain convinced that this is important work that should be published.

I am relieved to read that the authors are in agreement with my interpretation of ligand binding vs conformational changes for TG2. I hope my original comments were not too pedantic, but I do think it is important to be precise. By tightening the language with respect to how ligand binding relates to conformational changes, the authors' manuscript will make a very positive contribution to the literature. Hopefully, others in the field will gain a more nuanced understanding of the timing of conformational changes and ligand binding, thanks to the authors' careful description and sound interpretation of their distinctive data.

The authors also modified their manuscript to describe the mechanism of action of LM11 (with respect to conformational modulation) more precisely.

The original manuscript made some unsupported statements regarding the cellular permeability of different classes of inhibitors. This was apparently corrected by removing all such commentary. In fact, the modified manuscript presents very little medicinal chemistry at all, and this Reviewer wonders whether any 'pharmaceutically relevant properties' of the LM series of inhibitors were even cited in the manuscript, even though this phrase remains in the experimental section on page 23. This reviewer looks forward to future work from the authors, describing their medicinal chemistry in more detail, while acknowledging that removing that discussion does not detract at all from the focus of the current manuscript.

In summary, the authors have accounted for all of my suggested corrections, and I do believe the manuscript is even stronger now. I congratulate the authors for their fine work, and look forward to following its impact on the field.

On a final note, on looking through the references (just now) I noticed many duplications. (Sorry, I should have caught this in the initial review.) For example, reference 68 appears to be the same as 37. Reference 69 is identical to 50. 74 is the same as 54. There may be others I haven't noticed, but this needs to be cleaned up.

We appreciate the reviewers' continued support for our manuscript. We have updated the references to remove duplicates.